# Explaining the apparent impenetrable barrier to ultra-relativistic electrons in the outer Van Allen belt

Louis G. Ozeke[1], Ian R. Mann [1], Kyle R. Murphy[2], Alex W. Degeling[1,3], Seth G. Claudepierre[4] & Harlan E. Spence[5]

Recent observations have shown the existence of an apparent impenetrable barrier at the inner edge of the ultra-relativistic outer electron radiation belt. This apparent impenetrable barrier has not been explained. However, recent studies have suggested that fast loss, such as associated with scattering into the atmosphere from man-made very-low frequency transmissions, is required to limit the Earthward extent of the belt. Here we show that the steep flux gradient at the implied barrier location is instead explained as a natural consequence of ultra-low frequency wave radial diffusion. Contrary to earlier claims, sharp boundaries in fast loss processes at the barrier are not needed. Moreover, we show that penetration to the barrier can occur on the timescale of days rather than years as previously reported, with the Earthward extent of the belt being limited by the finite duration of strong solar wind driving, which can encompass only a single geomagnetic storm.

[1] Department of Physics, University of Alberta, Edmonton, Alberta T6G 2E1, Canada. [2] NASA Goddard Spaceflight Center, Code 674, Greenbelt, Maryland 20771, USA. [3] Institute of Space Science, Shandong University, Weihai, China. [4] Space Science Department, The Aerospace Corporation, El Segundo, California 90245, USA. [5] Institute for the Study of Earth, Oceans, and Space, University of New Hampshire, Durham, New Hampshire 03824-3525, USA. Correspondence and requests for materials should be addressed to L.G.O. (email: lozeke@ualberta.ca)

The radiation belts were one of the first discoveries of the space age. Early in-situ satellite observations revealed that the belts consist of relativistic electrons and ions, populating an inner and outer belt, which were separated by a region void of relativistic particles called the slot. Typically, ultra-relativistic electrons with energies $\gtrsim 2$ MeV are confined to the outer radiation belt[1, 2] outside the slot. It is generally agreed that the morphology and dynamics of the radiation belts are influenced by wave–particle interactions, which lead to both acceleration and loss. Acceleration processes, such as radial diffusion driven by ultra-low frequency (ULF) waves[3, 4] and/or local acceleration[5] through interaction with lower band whistler mode chorus[6], as well as wave–particle loss processes[7–12] must combine in such a way to create the observed belt dynamics. In addition, outward ULF wave radial transport in combination with magnetopause shadowing can also result in outer radiation belt electron loss[13–16]. However, how these processes act to collectively produce the observed morphology, and the relative importance of multiple competing acceleration and loss processes under different solar wind driving conditions, remain relatively poorly understood[5]. The NASA Van Allen Probes (formerly known as the Radiation Belt Storm Probes[17]) consist of two elliptically orbiting spacecraft with an inclination of ~ 10°, passing from altitudes of ~ 600 km at perigee to $L \sim 6$ at apogee (where $L$ is the distance from the equatorial crossing point of a dipole magnetic field line to the centre of the Earth, in units of Earth radii). Using ultra-relativistic ($\gtrsim 2$ MeV) electron measurements taken over the first 20 months of the Van Allen Probes mission, Baker et al.[18] identified a previously unknown and unexplained apparent impenetrable barrier located at $L \sim 2.8$, representing a point where the radial gradient of ultra-relativistic electron flux is extremely steep and beyond which the ultra-relativistic electrons do not penetrate. As described by Baker et al.[18], this apparent barrier is not co-located with the plasmapause or any other identifiable magnetospheric boundary. It has also recently been suggested that the sharp boundary in ultra-relativistic electron flux at $L = 2.8$ might be explained by losses from a sharp outer edge to fast pitch angle scattering into the atmosphere arising from a resonant interaction with man-made very-low frequency (VLF) waves injected into the magnetosphere by ground-based transmitters (e.g., Foster et al.[19]). Note, however, that at lower energies such a barrier feature is not as apparent, with lower energy electrons penetrating more frequently into the slot region[11, 20–22].

Here we present evidence that a truly impenetrable barrier at the inner edge of the outer ultra-relativistic radiation belt may not really exist. In addition, producing a steep drop off in the ultra-relativistic electron flux near $L \sim 2.8$ does not require a sharp outer edge to some efficient fast loss process. Instead, the apparent feature of an impenetrable barrier at a fixed location can be explained by dynamical variations in the rate of ULF wave inward radial diffusion. The feature of the apparent barrier is produced as a result of an effective time limit for strong driving, and hence the resulting finite interval of rapid inwards radial diffusion, which is imposed naturally by the temporal extent of the solar wind structures that produce geomagnetic storms. When the rate of ULF wave radial diffusion is properly quantified and the effects of the evolution of gradients in electron phase space density (PSD) are incorporated, the artefact of an apparent Earthward limit to ultra-relativistic electron transport in the outer Van Allen belt (i.e., the so-called impenetrable barrier) is naturally explained. An apparent barrier is also likely to form in other magnetised astrophysical plasma systems, where the dynamics of the system result from the action of a finite duration of enhanced astrophysical or stellar wind driving.

## Results

**ULF wave radial diffusion simulations.** The dynamics and energization of equatorially mirroring ultra-relativistic electrons in the outer radiation belt are simulated by solving the one-dimensional radial diffusion equation in a dipole magnetic field expressed in terms of the planetary geomagnetic activity index, Kp, the McIlwain $L$-shell[23], and time. The majority of current radiation belt modelling uses the empirical expression for the radial diffusion coefficient $D_{LL}$ given in Brautigam and Albert[24] (hereafter referred to as $D_{LL}$[B & A]). This is based on electromagnetic ULF waves and uses the approach outlined in Lanzerotti and Morgan[25] but is limited to statistics with a maximum Kp value of 6. More recently, Ozeke et al.[4] derived empirical expressions for $D_{LL}$ using the approached outlined in Fei et al.[26] based on the wave power in the azimuthal electric field of ULF waves, derived using data from ground-based magnetometers and also parameterised using Kp (hereafter referred to as $D_{LL}$[Ozeke]). In the results presented by Ozeke et al.[4] the statistics were also only shown to Kp = 6; however, the ground-based magnetometer ULF wave power statistics use a sufficiently long time period for that to be also extended to Kp values of 9 and these were presented in tabular form by Ozeke et al.[27] (see also Fig. 7 in Mann et al.[28]). Here we use the analytic expression for the radial diffusion coefficient from Ozeke et al.[4] to examine ULF wave transport. Note that the maximum Kp observed in the interval examined here is 7.7, which occurs for only one 3 h interval of Kp; there are also only a total of three 3 h intervals of Kp > 6 in the 20-month period examined by Baker et al.[18] and re-examined here. This justifies the use of the Ozeke et al.[4] expressions to higher Kp (see also Supplementary Fig. 1, which shows a comparison between the Ozeke et al.[4] analytic expression and Ozeke et al.[27] the statistics from for Kp = 6, 7, and 8).

For comparison the flux of ultra-relativistic electrons is simulated here using each of these two different expressions for the ULF wave radial diffusion coefficients. Note that in each of these definitions the diffusion coefficients are a strong function of Kp, such that changes in Kp, e.g., from ~ 2 to ~ 6 can increase both formulations of the diffusion coefficients by a factor of over 100. The resulting electron dynamics are shown in Fig. 1 from September 2012 to May 2014, the same interval examined in Baker et al.[18] (see Methods section and Supplementary Fig. 2 for further details).

In Fig. 1, the simulation results are compared to the flux observed by the Relativistic Electron Proton Telescope (REPT)[29] from the Energetic particle, Composition, and Thermal plasma (ECT) suite[30] on board the Van Allen Probes at energies of 3.4, 4.2, and 5.4 MeV. The simulation results shown in the third and fourth rows of Fig. 1 are derived using a series of first adiabatic invariant conserving simulations, which are then combined in an assumed dipole field to produce plots of the $L$- and time-dependent response of electron flux at fixed energy to match the native measurements from fixed energy channels from the REPT instrument. The simulation results clearly demonstrate the repeated creation of a very steep radial gradient in flux at the inner edge of the outer ultra-relativistic radiation belt, and that the location of this gradient changes with time and from storm to storm. However, for the entire period between September 2012 and May 2014 shown in Fig. 1, there is no evidence of penetration of ultra-relativistic electron flux inward of $L \sim 2.8$ in the simulation results. These modelling results are consistent with the observational results of Baker et al.[18] and demonstrate that despite a very steep flux profile at its edge, the apparently impenetrable barrier can be naturally explained by inwards ULF wave radial transport.

This result is largely independent of the choice of the statistical radial diffusion models assessed here, as the $D_{LL}$ values from

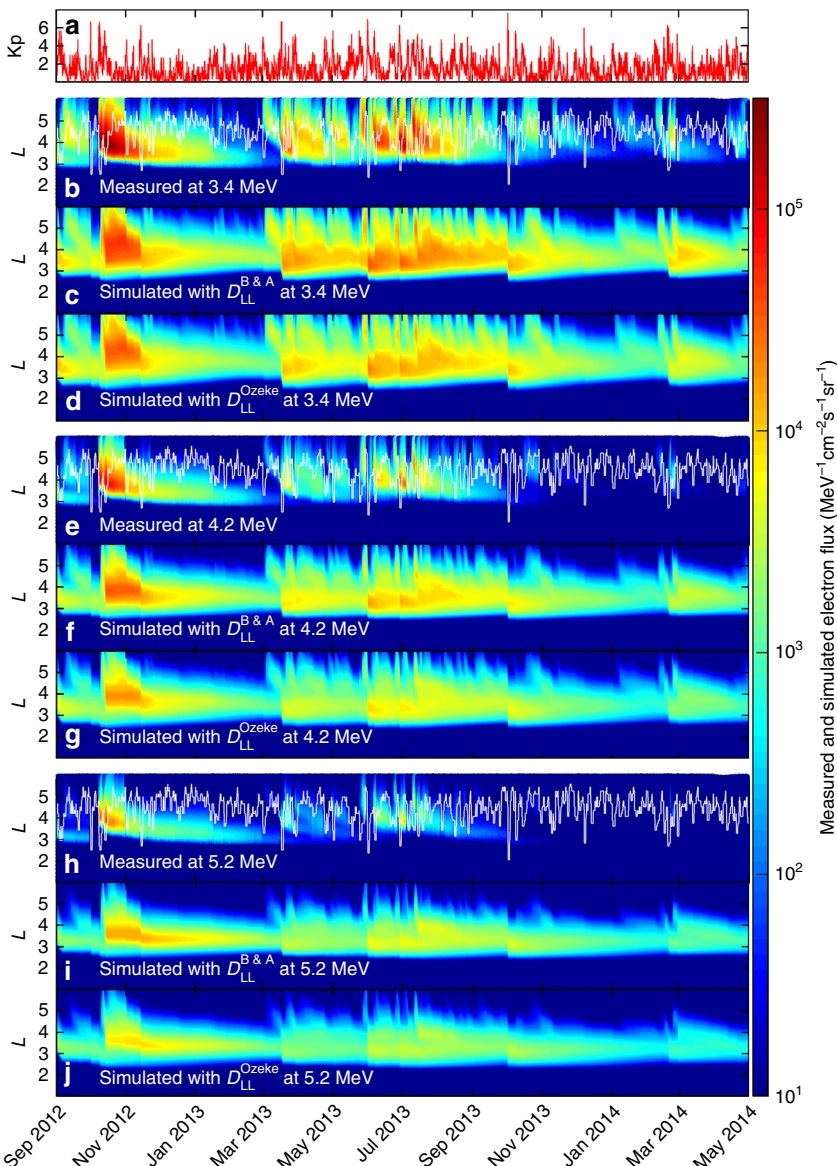

**Fig. 1** Measured and simulated ultra-relativistic electron flux from September 2012 to May 2014. The time series of Kp during this interval is shown in **a**. **b** The 3.4 MeV electron flux measured with the REPT instrument on Probes A and B during this time interval. **c, d** The simulated electron flux at a fixed energy of 3.4 MeV produced using a radial diffusion model applying the empirical radial diffusion coefficients from $D_{LL}$[B & A][24] and $D_{LL}$[Ozeke][4], respectively. Both these radial diffusion coefficient models are functions of the Kp values shown in **a**. Similar measured and simulated electron flux results are also shown at a fixed energy of 4.2 MeV in **e**, **f**, and **g**, and at a fixed energy of 5.2 MeV in **h**, **i**, and **j**. The location of the plasmapause derived from the Carpenter and Anderson[21] model is also over-plotted on the measured electron flux illustrated by the white curve in **b**, **e**, and **h**

both[24] ($D_{LL}$[B & A], Fig. 1c, f, i) and from[4] ($D_{LL}$[Ozeke], Fig. 1d, g, j) generate simulation results, which show a finite Earthward penetration that is limited to $L \sim 2.8$. In addition, the location of plasmapause shown as a white line in each of the panels b, e, and h, and derived from the Carpenter and Anderson model[31] does not correlate with the inner edge of the ultra-relativistic outer radiation belt. As pointed out by Baker et al.[18] and confirmed by our simulations, the apparent feature of an impenetrable barrier does not correspond to the location of the plasmapause or indeed to any other known physical boundary. Instead, and as we show here, the apparent barrier is naturally explained by the physics of ULF wave radial diffusion.

Overall, despite the use of Kp-dependent statistical representations of rates of ULF wave radial diffusion, the simulations show good agreement with the maximum depth of ultra-relativistic electron flux penetration as observed by the REPT instrument on

the Van Allen Probes. However, despite the good agreement with the ultra-relativistic electron penetration depth, the long-timescale dynamics of the absolute magnitude of the flux of ultra-relativistic electrons at all $L$-shells above the barrier is not explained perfectly by the one-dimensional dipole magnetic field model results shown in Fig. 1. For example, especially between $L = 3$ and $L = 5$, the simulated electron flux tends to be somewhat more intense than observed. Given the simplicity of both the one-dimensional model and the assumed dipolar geometry, and the neglect of flux changes arising from adiabatic effects from any time-dependence of the background magnetic field strength, absolutely perfect agreement is not to be expected and we discuss this further below. Nonetheless, the limit to the Earthward expansion of the belts is well-captured by the model.

To verify that the location of the apparent impenetrable barrier is explained by dynamic ULF wave radial diffusion transport,

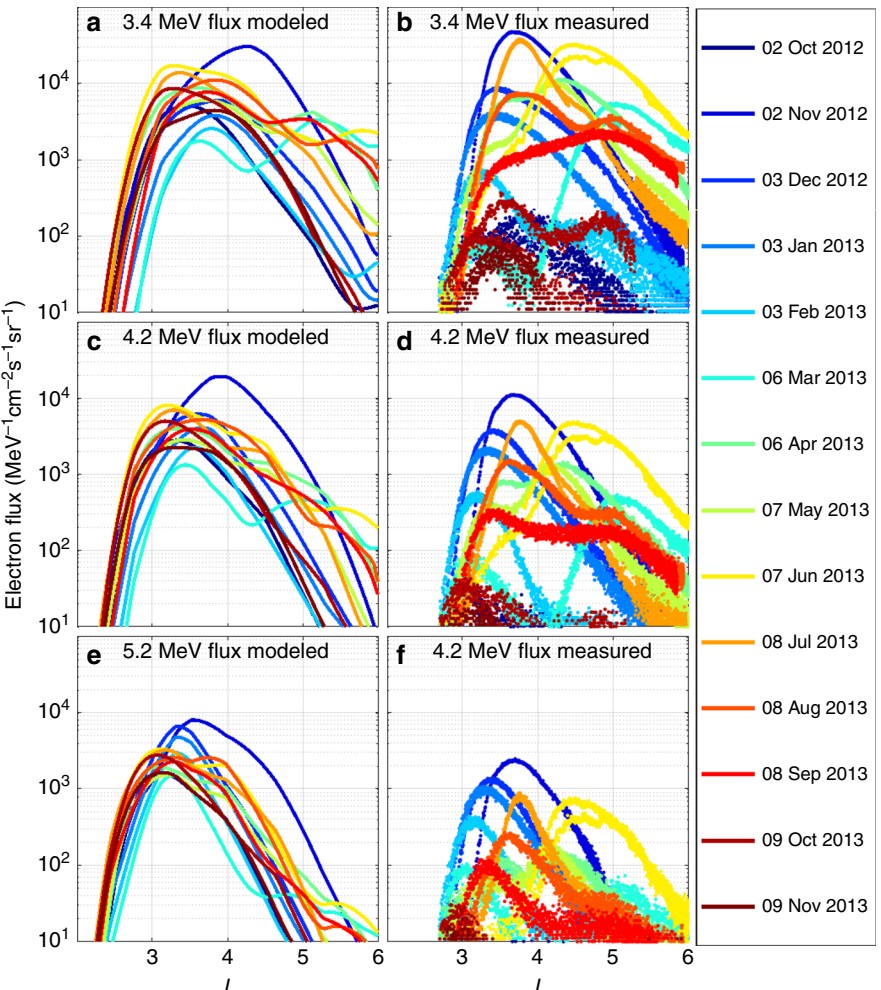

**Fig. 2** Observed and simulated ultra-relativistic electron flux radial profiles at time intervals of 31 days from 2 October 2012 to 9 November 2013. Each color-coded line represents a different profile of the differential electron flux at fixed energies of 3.4 MeV (**a**, **b**), 4.2 MeV (**c**, **d**), and 5.2 MeV (**e**, **f**). The panels on the left show the simulated electron flux profiles at 12 UT on each different day derived using the $D_{LL}$[Ozeke][4] diffusion coefficients. The panels on the right show the electron flux measured by Van Allen Probe A during each day from 0400 UT to 1600 UT

Fig. 2 shows a comparison between the differential flux in the modelled belt, calculated using $D_{LL}$[Ozeke] (left column), and that observed by the Van Allen Probes (right column), at fixed ultra-relativistic energies of 3.4, 4.2, and 5.2 MeV (top, middle, and bottom rows, respectively). $L$-dependent profiles are plotted every 31 days over the period from 2 October 2012 to 9 November 2013. The radial profiles of simulated electron flux were produced using initial electron flux values derived from measurements taken on 1 September 2012 and driven by outer boundary electron flux values derived from Van Allen Probe measurements taken at $L = 6$, using the technique described in the Methods section. The panels on the left of Fig. 2 show clearly how the simulation produces very steep gradients in the electron flux profile comprising the barrier at $L \sim 2.8$, very similar to those observed and shown in the panels on the right.

We emphasize that these simulations do not include any rapid electron loss mechanisms which are sharply confined to locations at or inward of $L \sim 2.8$. The steep electron flux profile at the inner edge simply results from the sharp falloff in the inward ULF wave diffusive transport rates with decreasing $L$-shell. These are of course coupled with slow electron loss due to wave-particle scattering losses to the atmosphere due to chorus and plasma-spheric hiss waves. It is interesting to note that even though the diffusion coefficients used in the simulations presented in both

Figs. 1 and 2 are varying with time by over three orders of magnitude, the location of inner-most edge of the ultra-relativistic outer radiation belt remains relativity constant at $L \sim 2.8$. As shown in Fig. 2, despite the fact that flux gradients can represent changes of four orders of magnitude in flux across $\sim 0.5$ $L$-shells, they are created and maintained naturally by the physics and the steep $L$-dependence of the rates of transport arising from ULF wave radial diffusion. Significantly, no sharp boundary in loss processes confined inside $L \sim 2.8$ is required.

Note that the results shown in Fig. 2 are generated from a set of single and continuous long duration (over 13 months) radial diffusion simulations at fixed adiabatic invariant such that the only constraints are the initial flux as a function of $L$ on 1 September 2012, the time series of the flux at the outer boundary at $L = 6$, and the empirical Kp-dependent rates of both ULF wave radial diffusion and chorus and hiss electron lifetimes. There is no update to any of the fluxes at lower $L$ inside the outer boundary through assimilation of observed flux. Hence, the dynamics of the system in Fig. 2 result predominantly from the inward/outward transport arising from the time-dependence of the flux observed by the Van Allen Probes at the outer simulation boundary coupled to lower $L$ by ULF wave radial diffusion. Note that the magnetic field is also assumed to be a dipolar for all time, such that short timescale effects from adiabatic changes in flux arising

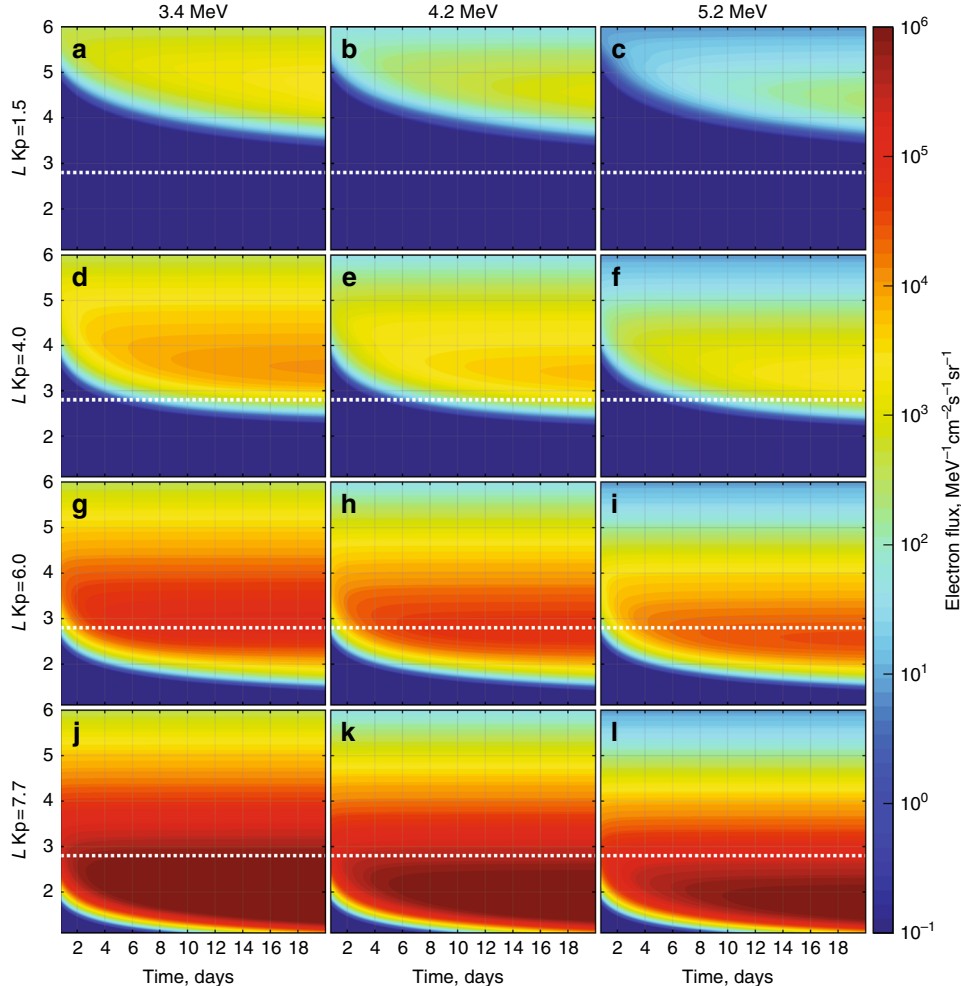

**Fig. 3** Radial diffusion simulations showing the inward progression of electron flux over a 20-day interval driven by radial diffusion coefficients specified by different fixed Kp-values. Each column shows the resulting flux at fixed energies of 3.4 MeV (left), 4.2 MeV (middle), and 5.2 MeV (right) arising from transport at fixed Kp of 1.5 (**a-c**), 4 (**d-f**), 6 (**g-i**), and 7.7 (**j-l**). The diffusion coefficients are specified by $D_{LL}$[Ozeke] and the initial electron flux is set to zero such that the belts are created from inward ULF wave transport. The electron flux at the outer boundary ($L = 6$) is specified using the mean electron flux at each energy channel measured by both Van Allen Probes A and B and derived from the entire period from 1 September 2012 to 31 May 2014. As is clearly shown, during active conditions the rates of radial transport from the outer simulation boundary to the heart of the radiation belt and beyond correspond to timescales on the order of a single magnetic storm

from time-dependent changes to the magnetic field in the equatorial plane of the magnetosphere are also not included in the model. Given the relative simplicity of the model, the existence and location of the apparent impenetrable barrier are well-explained as being the result of the inward extent of the ULF wave radial diffusion arising from ULF wave excitation in the Earth's magnetosphere by the solar wind.

The results shown in Figs. 1 and 2 do however show evidence for more variability in the flux in the heart of the radiation belt especially in terms of a larger loss over this period than is produced in our radial diffusion model. Given the statistical nature of the ULF wave diffusion coefficients and the representation of chorus and hiss loss rates, part of this discrepancy may result from uncertainties arising from the empirical Kp-dependent representation of the rates of ULF wave transport, and/or in the rates of loss due to hiss and chorus waves. In relation to the latter, the impact on the simulated flux of changing the hiss and chorus plasma wave-particle atmospheric scattering loss timescale is shown in Supplementary Fig. 2. This figure shows that the depth of electron penetration is not a strong function of the assumed hiss and chorus wave electron loss rates. This verifies

the validity of our result that the apparent barrier can be explained by ULF wave transport, and that this conclusion can be made largely independent of the exact rate of hiss and chorus loss for ultra-relativistic electrons. In relation to the former, in terms of the rates of ULF wave radial diffusion, Mann et al.[32] show that the observed ULF wave power and hence the rates of ULF wave transport can at times be much larger than predicted by the empirical Kp-dependence of statistical models. Indeed, as discussed by Mann and Ozeke[33] (see also Mann et al.[32]) ULF wave transport can be much faster than is typically assumed. For example, this can result in additional strong and rapid losses due to enhanced outwards radial diffusion to the magnetopause during the storm main phase leading to lower fluxes in the belts through what has been termed ULF wave enhanced magneto-pause shadowing (e.g., Mann et al.[32]). Indeed, Mann et al.[32] show that appropriate characterization of storm-time ULF wave power can also explain the generation of the third radiation belt reported by Baker et al.[34]. In our view, most likely this explains a very significant amount of the loss missing from Fig. 1.

In addition to chorus and hiss loss, there could also be additional losses arising from other wave modes such as

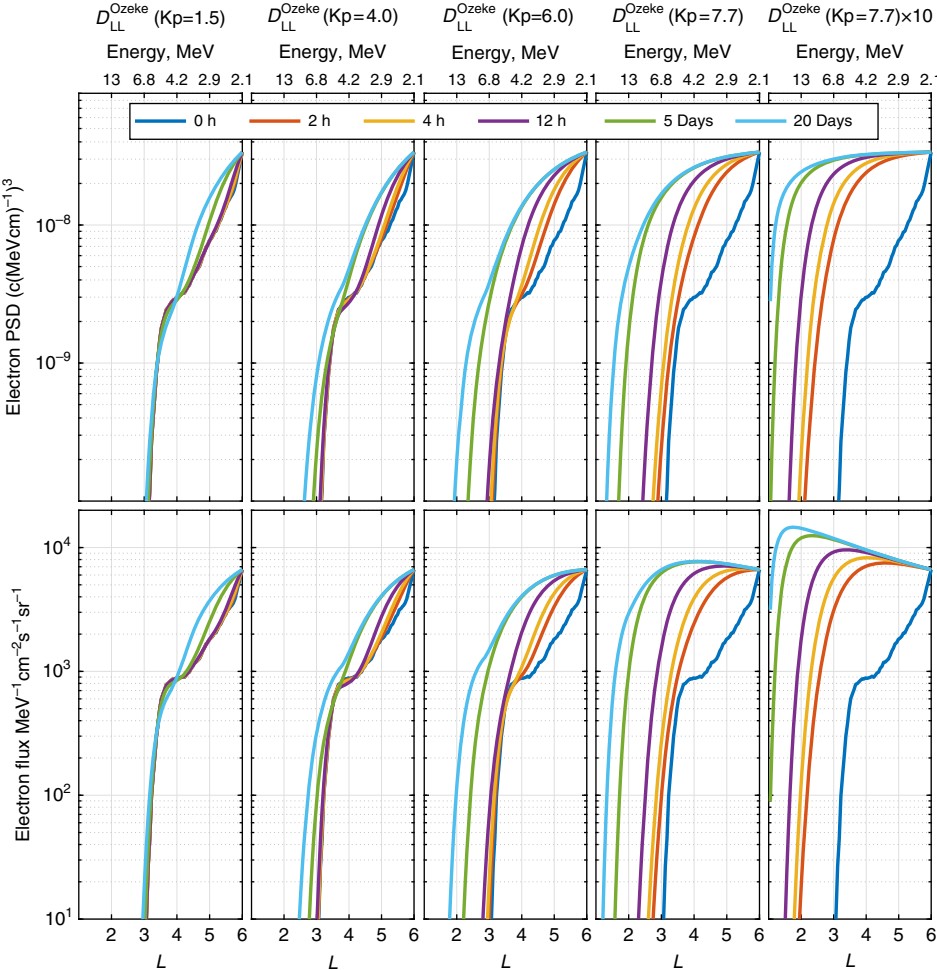

**Fig. 4** Ultra-relativistic electron phase space density (PSD) (top row) and flux (bottom row) profiles as a function of $L$-shell (bottom axes) and energy (top axes) for 1500 MeV/G electrons. The different colored curves indicate the simulated electron flux and PSD at different times, with the initial PSD and flux at $t = 0$ being derived from the flux measured by Van Allen Probe A on 1 September 2012. The flux and PSD at the outer boundary of $L = 6$ is derived from the mean 2.1 MeV electron flux measured by Van Allen Probes A and B from 1 September to 31 May 2014. The radial diffusion coefficients are specified using $D_{LL}$[Ozeke] at the fixed Kp-values indicated

electromagnetic ion cyclotron (EMIC) waves, which are neglected in this model. For example, Drozdov et al.[35] concluded that the effects of EMIC waves may be required to limit the long-term flux in the ultra-relativistic radiation belt. However, observations presented by Usanova et al.[36] show how the action of EMIC waves alone only impacts low equatorial pitch angle particles (where pitch angle is the angle between particle velocity and the background magnetic field) such that these waves acting alone are not expected to be able to deplete the core of the distribution[37, 38]. There could also be impacts from the action of chorus wave acceleration, as described, e.g., by Thorne et al.[6]. Although chorus waves may have an important role in the acceleration of electrons at relativistic energies[39, 40], at the ultra-relativistic energies ($\gtrsim 2$ MeV) examined here, the effects are expected to often be relatively weak[41]. Moreover, if additional chorus acceleration also primarily acts close to the outer boundary of our simulations the inward ULF wave transport of this additional source of flux will be captured in our model. Overall, and in spite of these provisos and the simplicity of our model, our results show that the apparent feature of an impenetrable barrier at the inner edge of the ultra-relativistic outer zone can be naturally and well-explained by the time-dependent action of inward ULF wave transport.

**On the existence of a truly impenetrable barrier**. In order to quantify the electron transport times as a function of Kp, the rates of dynamical penetration of ultra-relativistic electron flux at fixed energy under the action of ULF wave diffusion at fixed Kp from a constant outer boundary condition are shown in Fig. 3. Fig. 3 illustrates that, initially, the ultra-relativistic electron flux rapidly diffuses inward from the outer boundary. The minimum $L$-shell, which the electrons reach, depends on the strength of the diffusion coefficient (as specified over the extended intervals in Figs. 1 and 2 by Kp), the gradients which develop during the transport, and how long the enhanced radial diffusion lasts. The mean value of Kp over the 20 month time interval presented in Fig. 1 is 1.5 and the maximum value of Kp reached during the same interval was 7.7—the latter being maintained for only a single Kp resolution interval of 3 h.

The top panels a, b, and c of Fig. 3 illustrate that after 20 days of steady inward diffusion as specified by Kp = 1.5 the electron flux does not reach $L = 3$. For steady diffusion at Kp = 4 and Kp = 6 it takes ~ 10 days and ~ 2 days, respectively, for the effects of the enhanced flux at the outer boundary to reach $L = 2.8$. However, the bottom panels of Fig. 3 (which shows results for a constant Kp = 7.7, the largest value reached during the epoch shown in Fig. 1) illustrate that if an extended period of very fast

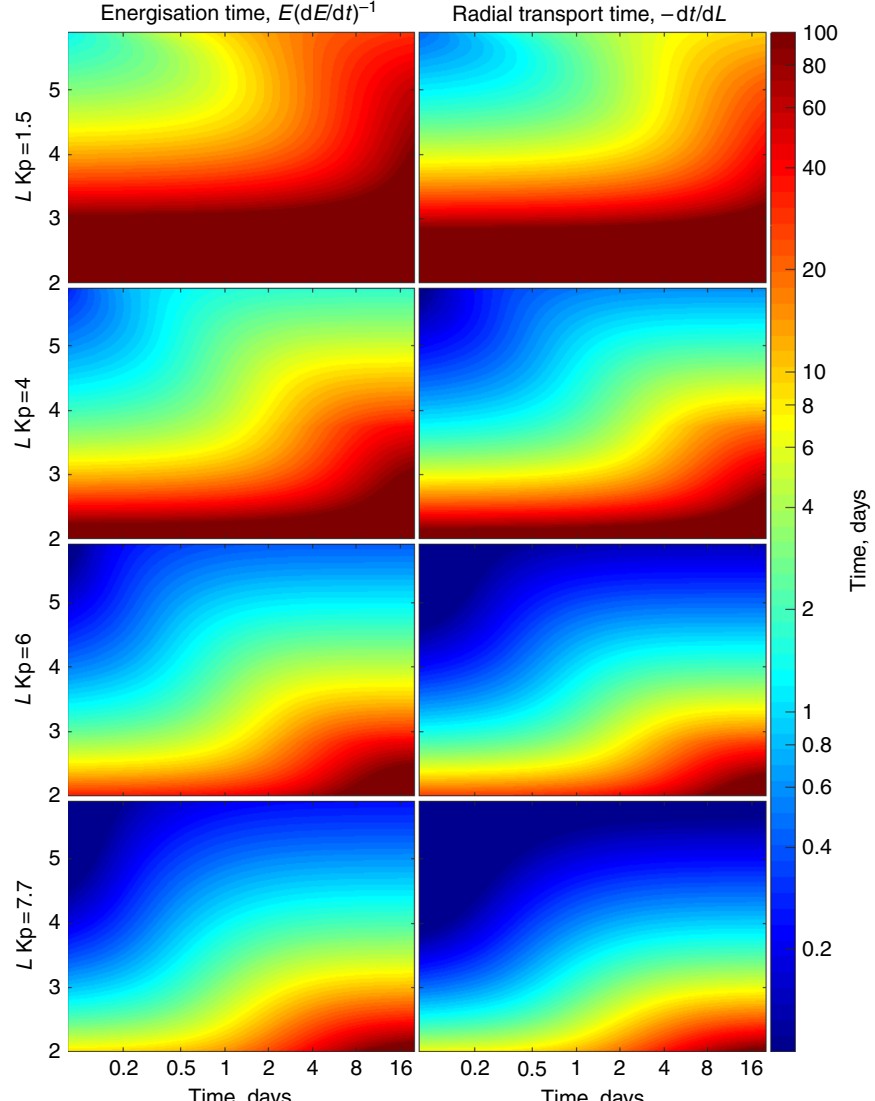

**Fig. 5** ULF wave radial diffusion energisation time and radial diffusion transport time as a function of *L*-shell and time. Color contours of the e-fold energization time specified by $E(dE/dt)^{-1}$ (left column, derived from Eq. (2)) and the radial transport time specified by $-(dL/dt)^{-1}$ (right column, derived from Eq. (1)) showing both of these times increase steeply with decreasing *L*-shell (see Methods section for more details). Similar to Fig. 4 in the main article, the initial electron PSD is set to zero, the PSD at the outer boundary is a constant derived from measured 2.1 MeV flux and the diffusion coefficients, $D_{LL}$[Ozeke] are specified by Kp = 1.5, 4.0, 6.0, and 7.7. Note that during the 20-month time interval examined by Baker et al.[18], at most, the geomagnetic activity remained at Kp > 1.5, Kp > 4.0, Kp > 6.0, and Kp = 7.7 for 126, 30, 9, and ≤ 3 h, respectively. As shown in this figure, these time intervals are all much shorter than the transport and energization times required to enhance the electron flux below *L* ~ 2.8 through ULF wave radial diffusion

transport persisted then the flux would in fact penetrate further Earthward than *L* ~ 2.8. For example, if the Kp = 7.7 conditions were to persist for ≳ 6 h then the ultra-relativistic electron flux would penetrate inward through the observed apparent barrier location at *L* ~ 2.8 (see also Supplementary Fig. 3, which shows these details over a shorter timescale). Moreover, if such high activity persisted for the unphysically long time of ≳ 2 days, the ultra-relativistic electron flux could reach locations below *L* = 2. Even longer intervals of very strong ULF wave activity at the level characteristic of Kp = 7.7 would drive the inward penetration even further. This shows that in fact a truly impenetrable barrier does not really exist and instead is an artefact of the magnitude and finite duration of the solar wind driving in producing rapid inward ULF wave radial diffusion. Indeed, as discussed, e.g., by Baker et al.[42], such penetrations of the barrier have previously been observed such as during the period of the 2003 Halloween

storms—with Loto'aniu et al.[43], suggesting that enhanced ULF wave power at low-*L* explain this penetration through ULF wave radial diffusion.

Both the observed and simulated electron flux in Fig. 1 also illustrate that in general enhancements in the electron flux correspond to sudden enhancements in the Kp index. This is consistent with the hypothesis that the electron flux enhancements result from rapid inward transport to low *L*-shells by ULF wave inward radial diffusion. In the case of the Earth's Van Allen belts, there is a finite duration of strong solar wind driving imposed by the scale of solar wind structures such as interplanetary coronal mass ejections and co-rotating interaction regions, which drive magnetic storms. Consequently, as a result of the finite duration of fast inwards transport, the depth of penetration of the ultra-relativistic belt is limited under typical storm conditions to be confined to *L* ~ 2.8 or higher, although

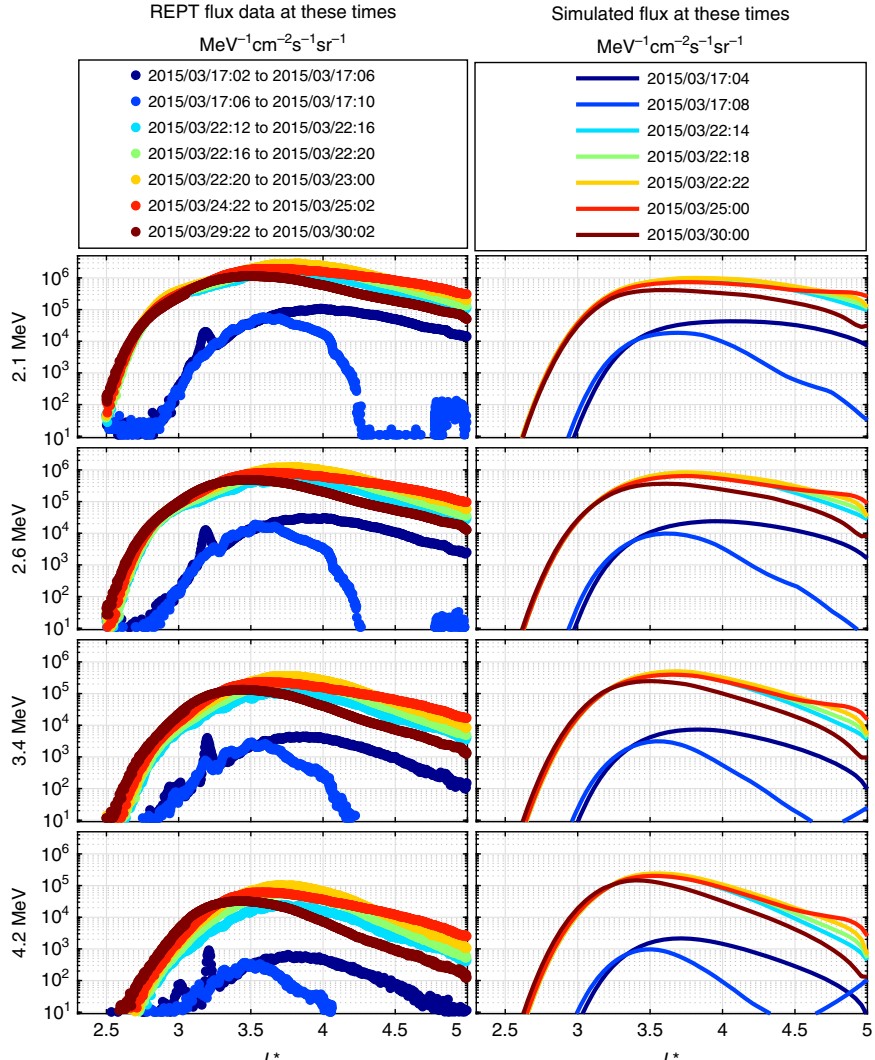

**Fig. 6** Apparent impenetrable barrier during the intense March 2015 geomagnetic storm. A comparison between the observed and simulated electron flux as a function of $L^*$ derived from the TS04D magnetic field model. The flux is illustrated at fixed energies of 2.1, 2.6, 3.4, and 4.2 MeV. The legend shows the times of the simulated and measured flux v $L^*$ profiles, from 17 March to 30 March 2015

during extreme events such as the Halloween 2003 storm it can occasionally penetrate deeper. The usual feature of a finite depth of penetration is maintained, even though the edge of this region is characterized by a very steep $L$-gradient in flux.

Similar to Fig. 2, Fig. 4 shows $L$-shell profiles of electron PSD and corresponding flux which further serve to illustrate how ULF wave radial diffusion naturally creates flux profiles which can have very steep gradients at lower $L$-values. For diffusion coefficients specified by Kp $\lesssim 6$, extended periods of driving of $\sim 2$–20 days are required for electron flux to be transported inward of $L = 2.8$. During periods characterized by Kp $> 6$, which are reflected in the Ozeke et al.[4] statistical model for ULF wave diffusion coefficients, it is possible for shorter intervals of driving to more rapidly transport electron flux to locations below $L = 2.8$. The last column on the right of Fig. 4 illustrates that when the diffusion coefficient is set artificially high to be 10 times greater than that specified by the average conditions characteristic of Kp $= 7.7$, then it is possible for the flux to be transported inward of even $L = 2$ on timescales of 4 h or less. Note, however, that as shown in Supplementary Fig. 1, the ULF wave power and hence the diffusion coefficients in the heart of the belt specified in statistical combinations of a power spectrum with a power law

plus a Gaussian enhancement presented by Ozeke et al.[27] can be higher by close to an order of magnitude than those in the analytic model of Ozeke et al.[4] (see, e.g., the case for Kp = 8 in Supplementary Fig. 1). Intense conditions such as these might occur during extreme geomagnetic superstorms and hence might be expected to be associated with penetration of the apparent barrier, as observed, e.g., during the Halloween 2003 superstorms[42] where two 24 h intervals of sustained activity with Kp > 6 occurred, see Supplementary Fig. 4). Under such conditions, the penetration of the barrier could allow flux to start to fill the slot region—so long as this extreme geomagnetic activity persists at an elevated level for sufficiently long time intervals (perhaps $\gtrsim 6$ h according to the results shown in the bottom row of Supplementary Fig. 3). Nonetheless, under typical solar wind driving conditions, the Earthward penetration of the electron flux is constrained to $L \gtrsim 2.8$ without requiring any sharp barrier or sharp onset of additional loss processes inside $L = 2.8$.

**Inward transport and energization timescales.** In examining their newly discovered feature of an apparent impenetrable barrier Baker et al.[18] stated "The radial transport of such electrons

from the heart of the outer zone to $L < 2.8$ is usually very slow (on the timescale of years). Thus, the electrons would be significantly depleted (by several orders of magnitude) by wave scattering during inward transport from the nominal plasmapause location at around four to five Earth radii". The argument that the electron transport timescales down to $L = 2.8$ are typically of the order of years presented by Baker et al.[18] is based on the radial diffusion model of Cornwall[44] who derived a diffusion coefficient model produced by substorm electrostatic fluctuations of the convection electric field. Part of the reason for the discrepancy between the Baker et al.[18] conclusion and the results presented here is that the Cornwall[44] model does not include electromagnetic fluctuations (with an induced electric field) which are much more efficient at transporting ultra-relativistic electrons; see, e.g., Brautigam and Albert[24]. In addition, the radial diffusion transport rate at any time is not only a function of the diffusion coefficient but also depends very strongly and dynamically on the electron PSD gradient, $\mathrm{d}f/\mathrm{d}L$. As such, the rates of radial diffusion cannot be derived solely from the diffusion coefficients by taking $D_{\mathrm{LL}}{}^{-1}$, as was done by Baker et al.[18]. Such an approach produces estimates of radial transport timescales, which are incorrect and unrealistically long.

The accurate way to express the effects of $\mathrm{d}f/\mathrm{d}L$ on the transport and energisation time of radiation belt electrons by ULF wave radial diffusion was given by Shultz and Lanzerotti[3], where the radial transport time $(\mathrm{d}L/\mathrm{d}t)^{-1}$ and the energisation timescale $E(\mathrm{d}E/\mathrm{d}t)^{-1}$ are given, respectively, by

$$\frac{\mathrm{d}L}{\mathrm{d}t} = -D_{\mathrm{LL}}\left(\frac{\partial \ln f}{\partial L}\right)_{M,J} \tag{1}$$

$$\frac{1}{E}\frac{\mathrm{d}E}{\mathrm{d}t} = \frac{1}{B_E}\frac{\mathrm{d}L}{\mathrm{d}t}\left[\frac{\gamma+1}{\gamma}\right]\frac{\partial B_E}{\partial L}. \tag{2}$$

Here, $\gamma$, $L$, and $B_E$ are the relativistic correction factor, $L$-shell, and the equatorial magnetic field at a given $L$-shell. Plots of this energization time and transport time are illustrated in Fig. 5, derived for fixed Kp-dependent diffusion coefficients $D_{\mathrm{LL}}$[Ozeke] specified by Kp = 1.5, 4.0, 6.0, and 7.7 (top to bottom rows, respectively). The time scale (color bar) represents the time taken for the electrons to move inward 1 Re and for the energy to experience an $e$-fold increase. Figure 5 illustrates clearly that the timescale for radial transport to reach the barrier does not equate to years but instead corresponds to the timescale of an individual storm; in addition, the transport and energization timescales shown in Fig. 5 are time-dependent verifying the fact that these timescales depend strongly on the local PSD gradient $\mathrm{d}f/\mathrm{d}L$—with these timescales getting slower as the system in Fig. 5 develops toward equilibrium. Of course, in the real magnetosphere changes to the flux at the outer boundary can hence be communicated inwards on the timescale of days (see also Mann and Ozeke[33]). Indeed, and as shown in Fig. 5, it is in fact quite possible for electrons to diffuse inward to $L = 2.8$ from $L = 6$ on geomagnetic storm timescales solely by the action of ULF wave radial diffusion. Therefore, and in contrast to the conclusion by Baker et al.[18], the explanation for the apparent impenetrable barrier is not "exceptionally slow natural inward radial diffusion combined with weak, but persistent, wave–particle pitch angle scattering" occurring on the "timescale of years" but instead can be explained as a result of the activity, time, and strong $L$-shell dependence of the rates of ULF wave radial diffusion during the course of a single magnetic storm. We demonstrate this further in the next subsection.

**Example of a strong geomagnetic storm in March 2015.** Very recently, Baker et al.[45] showed that even during the intense 17 March 2015 and 22 June 2015 storms the flux of these ultra-relativistic electrons did not significantly penetrate below the previously reported location of the impenetrable barrier at $L \simeq 2.8$. However, similar to the hypothesis of Foster et al.[19], the authors of Baker et al.[45] suggested that the impenetrable barrier is instead manmade and is produced by a bubble of VLF waves surrounding the Earth generated by ground-based radio transmitters in the 20–30 kHz frequency range.

To demonstrate that ULF wave transport can reproduce the characteristic feature of the barrier during a single storm, Fig. 6 shows the results from a radial diffusion simulation whereby observations from the REPT instrument (left column) are compared to those generated by the ULF wave radial diffusion model for the March 2015 storm (see, e.g., Baker et al.[45] and references therein). For this short interval, we show observational and model results as a function of $L^\star$ Roderer calculated from the TS04D model Tsyganenko and Sitnov[46]. Here, the values of the PSD at the outer simulation boundary at $L^\star = 5$ are derived from observed flux and the dynamic TS04D magnetic field. In addition, the statistical representations of the rates of diffusion are derived from observations from global ground-based magnetometer networks using the method described by Mann et al.[32]. In addition, the dipole $L$ expressions for the hiss and chorus losses, as well as the explicit dipole $L$-dependences in the diffusion coefficients, are also mapped dynamically from $L$ to $L^\star$ inside the simulation domain using the TS04D magnetic field model.

Figure 6 directly compares compare the $L^\star$ profiles of the measured ultra-relativistic electron flux with those obtained from our radial diffusion simulation during the March 2015 storm examined in Baker et al.[45], but does not include any effects from electron loss due to interactions with man-made VLF waves from transmitters. Both the measured and simulated electron flux $L^\star$ profiles before the storm on 17 March, indicated by the first two blue lines, are in remarkable agreement showing a rapid inner drop off in flux at $L^\star \simeq 3$. After the storm, and once the flux has reached an asymptotic inward location, as indicated by the solid lines 5, 8, and 13 days later, both the simulated and measured electron flux $L$-shell profiles show clear evidence for the barrier. At the inner edge, the flux profiles decrease by ~ 4 orders of magnitude in a spatial distance of only ~ 0.5 Re, reaching $L$-shells just below $L = 2.8$. This sharp feature, corresponding to the inner edge of the barrier, is produced primarily by the storm-time activity and $L$-dependence of the rates of radial diffusion.

Note that as discussed by Ozeke et al.[47] in relation to the extended radiation belt dropout interval in September 2014, during the main phase of magnetic storms there can often be a very rapid extinction of radiation belt flux on timescales shorter than the cadence provided by the orbit of the Van Allen probes and which can effectively wipe out the entire belt. Such extinctions can reduce the flux across the whole belt and effectively decouple the pre- and post-storm flux[47]. A similar radiation belt extinction to that reported by Ozeke et al.[47] for the September 2014 interval also occurs for the March 2015 storm (not shown, but see, e.g., Baker et al.[45] and Kanekal et al.[48] for details). The ULF wave radial diffusion simulation results presented in Fig. 6 hence assume that the flux is reduced to effectively zero across the whole belt on 17 March. Consequently, all of the flux in the post-storm period in Fig. 6 was created as the result of inward radial diffusion from the outer boundary following the extinction. Very significantly, this shows not only that ULF wave inward radial diffusion can explain the feature of the apparent impenetrable barrier but also that it is formed, and then remains in a fixed and stable location at $L \sim 2.8$, during a period of days during the course of a single magnetic storm.

Overall, the $L^*$ profiles of flux presented in Fig. 6 indicate that man-made VLF waves likely do not have a significant affect in producing the observed sharp ultra-relativistic electron flux drop off at $L \sim 2.8$ during the March 2015 storm. Combined with the additional long-timescale results shown above, our results support the hypothesis that the apparent impenetrable barrier is explained naturally as a result of ULF wave radial diffusion.

## Discussion

In their original paper, Baker et al.[18] suggested that ULF wave transport was too slow to enable electron flux to reach the location of the barrier. These authors hence suggested that local plasma wave-particle acceleration, such as might arise from resonance with chorus waves[6] was responsible for the ultra-relativistic electron flux reaching the location of the barrier. In addition, Tu et al.[49] also argue that local acceleration of electrons to multiple-MeV by strong chorus waves outside the plasmapause followed by slow inward radial diffusion may also explain the location of the barrier.

Here we presented evidence showing that ULF wave radial diffusion can transport the ultra-relativistic electron inward down to $L \sim 2.8$ consistent with the observed electron flux. Specifically, we show that the rates of ULF wave transport are both: (i) fast enough to rapidly transport electrons inward to the barrier during the period of the duration of a typical magnetic storm; (ii) slow enough once the storm abates to subsequently maintain the observed very steep flux gradient at the inner edge of the apparent barrier and hence effectively prevent any subsequent penetration further Earthward into the slot. Such an apparent barrier to ultra-relativistic radiation flux might also be expected in other astro-physical plasma systems perturbed aperiodically by a bursty stellar wind. If such systems have different characteristics, such an apparent barrier could however be located at a different radial distance from the magnetised body than in the terrestrial case.

## Methods

**ULF wave radial diffusion model.** The dynamics and energization of equatorially mirroring ultra-relativistic electrons in the outer radiation belt are simulated by solving the one dimensional radial diffusion equation in a dipole magnetic field expressed in terms of the McIlwain[23] $L$-shell, $L$, by Eq. (1).

$$\frac{\partial f}{\partial t} = L^2 \frac{\partial}{\partial L}\left[\frac{D_{LL}}{L^2}\frac{\partial f}{\partial L}\right] - \frac{f}{\tau}. \tag{3}$$

In Eq. (3), $f$ represents the PSD of the electrons and it is assumed that the first and second adiabatic invariants, $M$ and $J$, are conserved[3]. The radial diffusion coefficients and the electron lifetimes are represented by $D_{LL}$ and $\tau$, respectively. Both the initial electron PSD profile as a function of $L$ and the PSD at the outer boundary of the simulation (assumed fixed at $L = 6$) are derived from the Van Allen Probe measurements of the electron flux from both the MagEIS and REPT instruments[29,30,50]. Only MagEIS electron flux data was used from 37.3 keV to ~2 MeV and REPT data from 2.6 MeV to 5.2 MeV. At energies higher than 5.2 MeV, the REPT electron flux data at $L = 6$ appeared close to the instrument noise floor and the flux was hence assumed to be zero. Consequently, we can only simulate the electron flux upto energies of 5.2 MeV. Equation (1) was numerically solved for multiple first adiabatic invariants in order to simulate the electron flux, $J$, at a fixed energy, with the electron PSD converted to flux using the relationship

$$J = \frac{f}{p^2} \tag{4}$$

where $p$ is the relativistic momentum of an electron. Finally, at the inner boundary at $L = 1$, $f$ was effectively set to zero, representing loss to the atmosphere. The electron lifetime, $\tau$, outside and inside the plasmapause is defined using empirical representations based on the electron pitch angle scattering rates produced by chorus waves as presented in Gu et al.[10] and plasmaspheric hiss waves as presented in Orlova et al.[9]. Here the Carpenter and Anderson[31] model is used to specify the location of the plasmapause as a function of Kp. Our model does not include any local acceleration mechanisms such as those arising from lower band whistler mode chorus (e.g., Thorne et al.[6] and references therein), because these effects are relatively weak at ultra-relativistic energies[41] as compared with relativistic energies. Even if local acceleration rapidly creates an additional source for electrons around

$L = 5$ close to the edge of our simulations, as argued by Thorne et al.[6], the inward transport of such sources to the apparent barrier to ultra-relativistic electrons at $L \sim 2.8$ will also be captured in our simulations. Our model also does not include any pitch-angle scattering losses due to any waves other than chorus and plasmaspheric hiss. For example, magnetosonic and EMIC waves have been suggested as potential modes which may also enhance the electron loss (e.g., Shprits et al.[51] and Drozdov et al.[35]). No empirical expressions for the electron lifetimes as a function of $L$-shell due to magnetosonic or EMIC wave scattering into the atmosphere are currently available, and any potential loss effects from such waves are hence excluded in our simulations.

The electron flux profiles at fixed energy presented in Figs. 1–3 of the main article were derived from multiple first adiabatic invariant conserving simulations, with the electron flux at each $L$ being derived from a different first adiabatic invariant conserving run. In addition to the rates of transport, boundaries in flux can also be created by changes in the energy spectra at the outer boundary. For example, if the electron flux at the outer boundary drops off steeply with decreasing energy at some specific energy then if this source population is rapidly transported inward then the resulting profile of electron flux at a fixed energy will similarly have a steep decrease at some inner $L$-value as determined by the first adiabatic invariant. Consequently, the location of the inner edge of the outer radiation belt is controlled not only by the magnitude of the diffusion coefficients but also by both the dynamics and the energy spectrum of the flux at the outer boundary and the details of the radial gradients in $f$. In order to examine only the impact of radial diffusion on the location of the inner edge of the outer radiation belt and the time taken to reach the location of the apparent barrier, profiles of simulated electron flux, and corresponding PSD from a constant boundary condition at a fixed first adiabatic invariant for $D_{LL}$[Ozeke] radial diffusion coefficients constrained by fixed Kp were illustrated in Fig. 4 of the main article. In addition, plots of the energization and transport timescales are also presented in Fig. 5 of the main article using the approach defined by Schulz and Lanzerotti[3]. Finally, to demonstrate that the location of the apparently impenetrable barrier can be reached during the course of a single magnetic storm, ultra-relativistic electron transport due to radial diffusion driven by ULF waves was simulated for the intense March 2015 magnetic storm and the results shown in Fig. 6 of the main article. For that event, rather than using empirical characterizations as a function of Kp the ULF wave power levels were constrained by observations from ground-based magnetometers, the ground magnetic ULF power being mapped into electric field power in the equatorial plane (see, e.g., Mann et al.[32] and references therein for more details of the methodology).

**Data availability statement.** REPT and Magnetic Electron Ion Spectrometer data are available from the ECT suite on the Van Allen Probes (http://www.rbsp-ect.lanl.gov/) and geomagnetic indices from the World Data Center for Geomagnetism, Kyoto (http://wdc.kugi.kyoto-u.ac.jp/). All other data supporting the findings of this study are available from the authors upon request.

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

# ARTICLE

11. Ripoll, J. et al. Effects of whistler mode hiss waves in March 2013. *J. Geophys. Res. Space Phys.* **122**, 7433–7462 (2017).

12. Millan, R. & Thorne, R. Review of radiation belt relativistic electron losses. *J. Atmos. Sol. Terr. Phys.* **69**, 362–377 (2007).

13. West, H., Buck, R. & Walton, J. Shadowing of electron azimuthal-drift motions near the noon magnetopause. *Nature* **240**, 6–7 (1972).

14. Turner, D. L., Shprits, Y., Hartinger, M. & Angelopoulos, V. Explaining sudden losses of outer radiation belt electrons during geomagnetic storms. *Nat. Phys.* **8**, 208 (2012).

15. Ozeke, L. G. et al. Modeling cross L shell impacts of magnetopause shadowing and ULF wave radial diffusion in the Van Allen belts. *Geophys. Res. Lett.* **41**, 6556–6562 (2014).

16. Hudson, M. K. et al. Simulated magnetopause losses and Van Allen Probe flux dropouts. *Geophys. Res. Lett.* **41**, 1113–1118 (2014).

17. Kessel, R., Fox, N. & Weiss, M. The radiation belt storm probes (RBSP) and space weather. *Space Sci. Rev.* **179**, 531–543 (2013).

18. Baker, D. et al. An impenetrable barrier to ultrarelativistic electrons in the Van Allen radiation belts. *Nature* **515**, 531–534 (2014).

19. Foster, J. et al. Observations of the impenetrable barrier, the plasmapause, and the VLF bubble during the 17 March 2015 storm. *J. Geophys. Res. Space Phys.* **121**, 5537–5548 (2016).

20. Reeves, G. D. et al. Energy-dependent dynamics of keV to MeV electrons in the inner zone, outer zone, and slot regions. *J. Geophys. Res. Space Phys.* **121**, 397–412 (2016).

21. Ripoll, J. et al. Reproducing the observed energy-dependent structure of Earth's electron radiation belts during storm recovery with an event-specific diffusion model. *Geophys. Res. Lett.* **43**, 5616–5625 (2016).

22. Ma, Q. et al. Simulation of energy-dependent electron diffusion processes in the Earth's outer radiation belt. *J. Geophys. Res. Space Phys.* **121**, 4217–4231 (2016).

23. McIlwain, C. E. Co-ordinates for mapping the distributions of magnetically trapped particles. *J. Geophys. Res.* **66**, 3681–3691 (1961).

24. Brautigam, D. & Albert, J. Radial diffusion analysis of outer radiation belt electrons during the October 9, 1990, magnetic storm. *J. Geophys. Res. Space Phys.* **105**, 291–309 (2000).

25. Lanzerotti, L. & Morgan, C. G. ULF geomagnetic power near L=4: 2. Temporal variation of the radial diffusion coefficient for relativistic electrons. *J. Geophys. Res.* **78**, 4600–4610 (1973).

26. Fei, Y., Chan, A. A., Elkington, S. R. & Wiltberger, M. J. Radial diffusion and MHD particle simulations of relativistic electron transport by ULF waves in the September 1998 storm. *J. Geophy. Res. Space Phys.* **111** (2006).

27. Ozeke, L. G. et al. ULF wave derived radiation belt radial diffusion coefficients. *J. Geophys. Res. Space Phys.* **117**, A04222 (2012).

28. Mann, I. R. et al. The role of ultralow frequency waves in radiation belt dynamics. *Geophys. Monogr. Ser.* **199**, 69–91 (2012).

29. Baker, D. et al. The Relativistic Electron-Proton Telescope (REPT) instrument on board the Radiation Belt Storm Probes (RBSP) spacecraft: characterization of Earth's radiation belt high-energy particle populations. *Space Sci. Rev.* **179**, 337–381 (2013).

30. Spence, H. E. et al. Science goals and overview of the radiation belt storm probes (RBSP) energetic particle, composition, and thermal plasma (ECT) suite on NASA's Van Allen probes mission. *Space Sci. Rev.* **179**, 311–336 (2013).

31. Carpenter, D. & Anderson, R. An ISEE/whistler model of equatorial electron density in the magnetosphere. *J. Geophys. Res. Space Phys.* **97**, 1097–1108 (1992).

32. Mann, I. R. et al. Explaining the dynamics of the ultra-relativistic third Van Allen radiation belt. *Nat. Phys.* **12**, 978–983 (2016).

33. Mann, I. R. & Ozeke, L. G. How quickly, how deeply, and how strongly can dynamical outer boundary conditions impact Van Allen radiation belt morphology?. *J. Geophys. Res. Space Phys.* **121**, 2016JA022647 (2016).

34. Baker, D. N. et al. A long-lived relativistic electron storage ring embedded in Earth's outer Van Allen belt. *Science* **340**, 186–190 (2013).

35. Drozdov, A. et al. Energetic, relativistic, and ultrarelativistic electrons: comparison of long-term VERB code simulations with Van Allen Probes measurements. *J. Geophys. Res. Space Phys.* **120**, 3574–3587 (2015).

36. Usanova, M. et al. Effect of EMIC waves on relativistic and ultrarelativistic electron populations: ground-based and Van Allen Probes observations. *Geophys. Res. Lett.* **41**, 1375–1381 (2014).

37. Li, W., Shprits, Y. & Thorne, R. Dynamic evolution of energetic outer zone electrons due to wave-particle interactions during storms. *J. Geophys. Res. Space Phys.* **112**, A03208 (2007).

38. Lyons, L. R. & Thorne, R. M. Parasitic pitch angle diffusion of radiation belt particles by ion cyclotron waves. *J. Geophys. Res. Space Phys.* **77**, 5608–5616 (1972).

39. Boyd, A. J. et al. Quantifying the radiation belt seed population in the 17 March 2013 electron acceleration event. *Geophys. Res. Lett.* **41**, 2275–2281 (2014).

40. Li, W. et al. Radiation belt electron acceleration by chorus waves during the 17 March 2013 storm. *J. Geophys. Res. Space Phys.* **119**, 4681–4693 (2014).

41. Shprits, Y. Y. et al. Unusual stable trapping of the ultrarelativistic electrons in the Van Allen radiation belts. *Nat. Phys.* **9**, 699 (2013).

42. Baker, D. et al. An extreme distortion of the Van Allen belt arising from the/ Hallowe'en/'solar storm in 2003. *Nature* **432**, 878–881 (2004).

43. Loto'Aniu, T. M. et al. Radial diffusion of relativistic electrons into the radiation belt slot region during the 2003 Halloween geomagnetic storms. *J. Geophys. Res.* doi: https://doi.org/10.1029/2005JA011355 (2006).

44. Cornwall, J. M. Diffusion processes influenced by conjugate-point wave phenomena. *Radio Sci.* **3**, 740–744 (1968).

45. Baker, D. N. et al. Highly relativistic radiation belt electron acceleration, transport, and loss: large solar storm events of March and June 2015. *J. Geophys. Res. Space Phys.* **121**, 6647–6660 (2016).

46. Tsyganenko, N. & Sitnov, M. Modeling the dynamics of the inner magnetosphere during strong geomagnetic storms. *J. Geophys. Res. Space Phys.* **110**, A03208 (2005).

47. Ozeke, L. G., Mann, I. R., Murphy, K. R., Sibeck, D. G. & Baker, D. N. Ultra-relativistic radiation belt extinction and ULF wave radial diffusion: Modeling the September 2014 extended dropout event. *Geophys. Res. Lett.* **44**, 2624–2633 (2017).

48. Kanekal, S. et al. Prompt acceleration of magnetospheric electrons to ultrarelativistic energies by the 17 March 2015 interplanetary shock. *J. Geophys. Res. Space Phys.* **121**, 7622–7635 (2016).

49. Tu, W. et al. Modeling the impenetrable barrier to inward transport of ultra-relativistic radiation belt electrons (AGU Fall Meeting Abstracts, 2014).

50. Blake, J. et al. The magnetic electron ion spectrometer (MagEIS) instruments aboard the radiation belt storm probes (RBSP) spacecraft. *Space Sci. Rev.* **179**, 383–421 (2013).

51. Shprits, Y. Y. et al. Wave-induced loss of ultra-relativistic electrons in the Van Allen radiation belts. *Nat. Commun.* **7**, 12883 (2016).

## Acknowledgements

I.R.M. is supported by a Discovery Grant from Canadian NSERC. This work is supported by the Canadian Space Agency through the Geospace Observatory (GO) Canada program. K.R.M. was supported by an NSERC Postdoctoral Fellowship. We acknowledge the WDC for Geomagnetism, Kyoto University, Japan for the geomagnetic indices. This work was supported by RBSP-ECT funding provided by JHU/APL Contract No. 967399 under NASA's Prime Contract No. NAS5-01072. The authors thank Prof. Daniel N. Baker for useful discussions, and thank him and the entire REPT team for data. L.G.O also thanks A. Kale for formating the figures.

## Author contributions

L.G.O. and I.R.M. conceived the research and wrote the manuscript, which all authors read and commented on. K.R.M., S.G.C., and H.E.S. provided and helped analyse the data used in the study. L.G.O. and A.W.D. developed the code used to perform the electron flux simulations.

## Additional information

**Competing interests:** The authors declare no competing interests.

