## [Peer Review File · Nature Communications]

Reviewers' comments:

Reviewer #1 (Remarks to the Author):

The major claim of this paper is that the apparent impenetrable barrier to ultra-relativistic electrons in Earth's inner magnetosphere can be explained by ULF wave-driven radial diffusion.

When in 2014 an "impenetrable barrier" to ultra-relativistic electrons was reported to exist, this referee was highly sceptical that such a barrier actually exists. Moreover, that man-made VLF transmissions were required to explain a barrier that did not exist made less sense to this referee. To this referee, taking into account well known modes of plasma transport in Earth's inner magnetosphere, the simple explanation of any innermost location of highly energetic electrons was likely to involve (ULF wave driven) radial diffusion, namely inward radial diffusion simply "running out of steam" at some point, the exact spatial location depending on the strength of the wave-driven diffusion. The present submitted paper quantifies this referee's expectation.

Clearly, the present paper deserves to be published as it brings common sense, and, one would hope, closure to the so-called impenetrable barrier issue.

However, before I recommend publication, I would like the authors to reference and discuss the Abstract by W. Tu et al, the oral presentation being given at the Fall AGU Meeting 2014, Abstract # SM32A-03, entitled, Modeling the impenetrable barrier to inward transport of ultra-relativistic radiation belt electrons

This paper uses slow inward diffusion as a key ingredient in the explanation of the "barrier", and as such I think fully deserves to be referenced and discussed in the submitted article by Ozeke et al.

Reviewer #2 (Remarks to the Author):

Overall assessment

* The major claim of the manuscript under review is that the inability of highly relativistic electrons to penetrate much inside $L = 2.8$, even for highly disturbed conditions, can be simply explained as a consequence of the radial diffusion rates due to ULF waves.

* The authors take issue with some statements made in the Nature paper of Baker et al. [2014], which first described the phenomena of an impenetrable barrier, but their overall claim is basically similar. Namely that inward diffusion during even during large magnetic storms is not fast enough to carry relativistic electrons inward much past $L = 2.8$. The novelty in the current approach is their modeling of inward diffusion under a variety of conditions using a new form for the rate of radial diffusion based on the analysis of Ozeke et al [2014].

* Unfortunately, the current modeling performed is not state of the art, since it ignores several key physical processes such as local energy diffusion and various loss processes. As such I do not find it convincing.

* The paper would be made more convincing if these other known processes, which have now been accepted by most of the magnetospheric community, were included in the modeling. It should be a relatively easy task to include realistic loss by a combination of hiss and chorus since these have been parameterized by K_p , L and electron energy in a variety of recent publications. Inclusion of the effect of local energy diffusion will be more difficult since this process produces peaks in phase space density, which control the direction (inward or outward) of radial diffusion, but this has also been modeled by other groups before.

*Referencing of important related earlier work is sadly deficient, and the reference list is heavily slanted to previous studies of the Calgary group. This needs to be seriously addressed in any resubmission. The paper is also very long and contains material, which does little to support the main claim.

* Overall, I agree with the authors that the primary factor contributing to the inability of highly relativistic electrons to penetrate inward of $L= 2.8$ is the extremely slow inward transport rates and the limited duration of faster solar wind driving during most storms. However, this basic concept dates back to the 1970's [Lyons and Thorne, 1973] and is not in itself a new idea. Nonetheless, the paper would still be of interest to the community if the diffusion modeling were made more convincing and I would recommend that the authors consider a resubmission after considering the comments below.

Specific comments for the authors to address

P2 There has been substantial improvement in our quantitative understanding of the relative importance of various processes that affect the radiation belts since the paper of Reeves et al. [2013]. The statement that the relative importance of these processes "is not well understood" is not correct.

P2 The apparent barrier only applies to the highly relativistic population. Lower energy electrons are able to be transported into lower L during solar activity and fill the gap between the inner and outer radiation belts [see Ma et al., 2016]. This is an important distinction that needs to be discussed in the concept of energy dependent radial transport.

Section on Results: In Fig 1 there appears to be little difference in the modeling results using the radial diffusion rates of B&A and Ozeke, and neither of the two simulations are able to explain the apparently slow (months) inward diffusion of 4.2 and 5.6 MeV electrons following a major storm. This suggests that neither radial diffusion rate is correct during less active times following a storm.

Setting an outer boundary condition from observed flux measurements at $L=6$ is not appropriate since this ignores the possibility of any local acceleration source, which has been shown to produce peaks in phase space density between $L = 4$ to 5 during storms. When such peaks are formed the radial diffusion direction is outward to regions of lower PSD near the boundary [Turner et al., 2014]. This leads to the concept of magnetopause shadowing as a loss process for the radiation belts, which has been addressed in numerous papers [e.g., Shpritz et al., 2006] prior to the referenced paper of Mann et al., [2016]. This is but one example of the biased choice of references used in the manuscript.

Another example of misleading referencing occurs in the discussion of losses due to EMIC waves. Here the paper of Usanova et al [20014] is cited to explain the fact that EMIC waves are only responsible for scattering of electrons at low pitch angles around the loss cone. But this was clearly shown in much earlier studies [e.g., Lyons and Thorne, 1972; Li et al., 2007].

Section on "Does the Barrier exist".

In this section the authors use their model for radial diffusion to simulate the potential effect of strong solar wind driving over an extended period of time to show that radial diffusion could bring relativistic electrons into lower L if high levels of activity ($Kp > 6$) were sustained for tens of days. However, this is very unrealistic and has never been observed in Nature. Consequently, I would suggest removing this section or at least moving it into supplementary material.

Section on March 2015 storm

I agree with the authors that rapid loss from VLF transmitters is not needed to explain the sharp inner gradient in PSD formed during this particular storm. But here again the simulations performed to support this do not contain any realistic losses by wave scattering or the effect of local energy diffusion outside the plasmopause during interactions with chorus waves. It would be much more convincing if such effects were included.

Reviewer #3 (Remarks to the Author):

This paper presents very interesting results showing the so-called 'impenetrable barrier' at $L=2.8$ (Baker et al.) for highly relativistic electrons can be explained by the properties of radial diffusion due to ULF waves, in combination with the typical time-scales for large storms. The authors do a careful job of showing the effects of strong solar wind driving (as parameterized by K_p) on the timescale for relativistic electrons to reach different L values. They show that in fact, for larger driving, ultra-relativistic electrons can penetrate inside $L=2.8$ and that there is nothing special about that location.

As the authors point out, a number of other researchers have shown that during very large storms electrons are seen inside this location.

Issues to be addressed before publication:

The referencing of other work on ULF waves impact on radiation belts is not complete.

Although loss due to whistler-mode waves is addressed, other loss mechanisms such as magnetopause shadowing are not addressed.

Figure 1 should be redrafted so that each energy is full page width. It is not possible to see the relationship between the plasmopause location and the electron fluxes on the scale shown.

The figure caption is also confusing and should be .

This is true of several other figure captions, which do not stand on their own. One can't interpret the figure from the captions alone.

The authors do not need to continually emphasize the term used in the Baker et al. (and following articles) paper - "impenetrable barrier" - through use of italics and quotation marks.

We thank reviewer #1 for the careful reading of our manuscript. We have revised the paper taking into account the comments from all of the reviewers in our revision.

In our responses below, we address the comments from referee #1 with the original referee comments presented in **bold typeface** and our replies in regular font.

We hope that with these changes the referee can now recommend our paper for publication in Nature Communications.

Reviewer #1

The major claim of this paper is that the apparent impenetrable barrier to ultra-relativistic electrons in Earth's inner magnetosphere can be explained by ULF wave-driven radial diffusion.

When in 2014 an "impenetrable barrier" to ultra-relativistic electrons was reported to exist, this referee was highly sceptical that such a barrier actually exists. Moreover, that man-made VLF transmissions were required to explain a barrier that did not exist made less sense to this referee. To this referee, taking into account well known modes of plasma transport in Earth's inner magnetosphere, the simple explanation of any innermost location of highly energetic electrons was likely to involve (ULF wave driven) radial diffusion, namely inward radial diffusion simply "running out of steam" at some point, the exact spatial location depending on the strength of the wave-driven diffusion. The present submitted paper quantifies this referee's expectation.

Thank you.

Clearly, the present paper deserves to be published as it brings common sense, and, one would hope, closure to the so-called impenetrable barrier issue.

Thank you. With the publication of this paper we obviously hope so too.

However, before I recommend publication, I would like the authors to reference and discuss the Abstract by W.Tu et al, the oral presentation being given at the Fall AGU Meeting 2014, Abstract # SM32A-03, entitled, Modeling the impenetrable barrier to inward transport of ultra-relativistic radiation belt electrons.

This paper uses slow inward diffusion as a key ingredient in the explanation of the "barrier", and as such I think fully deserves to be referenced and discussed in the submitted article by Ozeke et al.

Overall we are very pleased that the referee thinks *"the present paper deserves to be published as it brings common sense, and, one would hope, closure to the so-called impenetrable barrier*

issue". As requested by the reviewer, in the revised manuscript we now include a reference to the oral presentation given at fall the AGU Meeting 2014, Abstract # SM32A-03, entitled "*Modeling the impenetrable barrier to inward transport of ultra-relativistic radiation belt electrons*" as well as a brief discussion of these results. At the end of the first paragraph in the Discussion and Conclusion section the revised manuscript now states:

"In addition, Tu et al.⁴⁴ also argue that local acceleration of electrons to multiple-MeV by strong chorus waves outside the plasmopause followed by slow inward radial diffusion may also explain the location of the barrier."

Reply to Referee #2

We thank the reviewers for their careful reading of our manuscript. We have revised the paper taking into account the comments from all of the reviewers in our revision. In our responses below, we address the original comments from Referee #2 which are presented in **bold typeface** and present our replies in regular font.

We hope that with these changes referee #2 can now recommend our paper for publication in Nature Communications.

Reviewer #2 (Remarks to the Author):

Overall assessment

*** The major claim of the manuscript under review is that the inability of highly relativistic electrons to penetrate much inside $L=2.8$, even for highly disturbed conditions, can be simply explained as a consequence of the radial diffusion rates due to ULF waves.**

Thank you. Yes, we believe that this is clearly demonstrated in our revised paper and offers an important and significant explanation of the unexpected Barrier first reported by in Nature by Baker et al. [2014].

*** The authors take issue with some statements made in the Nature paper of Baker et al. [2014], which first described the phenomena of an impenetrable barrier, but their overall claim is basically similar.**

As we describe in detail in our revised paper, the explanation offered by Baker et al. required local plasma wave-particle acceleration, such as might arise from resonance with chorus waves close to the plasmopause. Such a local source of ultra-relativistic electrons relatively proximal to the barrier was needed since Baker et al. claimed that radial transport was too slow stating that *“The radial transport of such electrons from the heart of the outer zone to $L<2.8$ is usually very slow (on the timescale of years).”* (direct quote).

In contrast to the conclusion of Baker et al., we show how more appropriately characterised ULF wave transport rates can enable the barrier to be reached during the course of a single storm (on a timescale of days, rather than years). In addition, no local acceleration effects are needed. But despite the huge discrepancy in timescale, and the fact that local acceleration is not required, the referee is correct that the basic concept of how the barrier is reached is in essence somewhat similar to that proposed by Baker et al.

Namely that inward diffusion during even during large magnetic storms is not fast enough to carry relativistic electrons inward much past $L=2.8$.

This is true. However, our modelling shows clearly that (partly due to the very steep L-dependence of the diffusion coefficient) the inward ULF wave diffusion (to use the expression used by Referee #1) “*runs out of steam*” during the course of a single storm and that this explains the location and existence of the apparent barrier. As explained above, this aspect is very different from the explanation advanced by Baker et al.

The novelty in the current approach is their modeling of inward diffusion under a variety of conditions using a new form for the rate of radial diffusion based on the analysis of Ozeke et al [2014].

Yes. Thank you. In addition, we would argue not only is radial diffusion the explanation, but also (as discussed in our response above) the time scales and fundamental basis of how the barrier is created is different from that proposed by Baker et al. Indeed, no modelling of radial diffusion is presented in Baker et al. – here we present details of such modelling and show how this provides a quantitative explanation for the existence and location of the barrier.

***Unfortunately, the current modeling performed is not state of the art, since it ignores several key physical processes such as local energy diffusion and various loss processes. As such I do not find it convincing.**

***The paper would be made more convincing if these other known processes, which have now been accepted by most of the magnetospheric community, were included in the modeling.**

It should be a relatively easy task to include realistic loss by a combination of hiss and chorus since these have been parameterized by Kp, L and electron energy in a variety of recent publications.

We apologise to the referee if our original paper was not clear, but the simulation results presented already do include the effects of realistic wave-particle losses as requested by the reviewer. This is explained very clearly in the Methods section of the manuscript, which is the style and section formatting adopted in Nature journals.

Inclusion of the effect of local energy diffusion will be more difficult since this process produces peaks in phase space density, which control the direction (inward or outward) of radial diffusion, but this has also been modeled by other groups before.

The referee is correct that including these energy diffusion effects would indeed be more challenging. However, fortunately the exclusion of these local energy diffusion effects does not invalidate our results as these effects are completely negligible at ultra-relativistic energies.

For example as discussed by Shprits et al. (2013) in their Nature Physics paper, “*diffusive local acceleration, which is very efficient at energies below 2 MeV, becomes inefficient for such high-energy [ultra-relativistic] electrons.*” (direct quote). To make this clearer to the reader we have now included additional text to make this point: on page 28 we state “Our model does not include any local acceleration mechanisms such as those arising from lower band whistler mode

chorus (e.g., Thorne et al.⁶ and references therein) not least because these effects are very weak at ultra-relativistic energies⁴⁶, and does not include any pitch-angle scattering losses due to any waves other than chorus and plasmaspheric hiss”.

In the text we also add a new reference to the Shprits et al. (2013) work quoted above: Shprits, Yuri Y., Dmitriy Subbotin, Alexander Drozdov, Maria E. Usanova, Adam Kellerman, Ksenia Orlova, Daniel N. Baker, Drew L. Turner, and Kyung-Chan Kim. "Unusual stable trapping of the ultrarelativistic electrons in the Van Allen radiation belts." *Nature Physics* 9, no. 11 (2013): 699.

The barrier is also observed most clearly at ultra-relativistic energies, and the title of our paper makes it clear that we are seeking to simulate the response at ultra-relativistic energies. Of course if we were examining dynamics at lower energies, energy diffusion effects might be important. But at ultra-relativistic energies energy diffusion effects can be neglected.

With the explanation above, we hope the referee will then find our results convincing.

***Referencing of important related earlier work is sadly deficient, and the reference list is heavily slanted to previous studies of the Calgary group. This needs to be seriously addressed in any resubmission. The paper is also very long and contains material, which does little to support the main claim.**

We apologise that the reviewer found our reference list to be deficient. No specific additional references are requested at this point in the referee’s report. However, in order to take the request that this be “*seriously addressed*”, our revision includes additional references requested by the referee in later comments below. In relation to radial diffusion we also add a reference to the seminal work on particle diffusion in the radiation belts by Schulz and Lanzerotti, originally from 1974 and republished in 2012:

Schulz, M. & Lanzerotti, L. J. in *Particle diffusion in the radiation belts* (Springer Science & Business Media, 2012).

Note that our manuscript has no authors from the University of Calgary or the “Calgary Group”.

*** Overall, I agree with the authors that the primary factor contributing to the inability of highly relativistic electrons to penetrate inward of $L=2.8$ is the extremely slow inward transport rates and the limited duration of faster solar wind driving during most storms.**

Thank you. To our knowledge there are no papers in the literature which provide quantitative modeling which shows that this is the case, and our paper hence aims to rectify that here with publication of our paper in *Nature Communications*.

However, as noted above, our emphasis (in contrast to Baker et al.) is that the transport happens on the timescale of a single storm – and hence significantly this transport is not “*extremely slow*” as claimed Baker et al. As mentioned above, Baker et al. quote a diffusive “*timescale of years*” in their paper, rather the days of a geomagnetic storm duration we show here.

However, this basic concept dates back to the 1970's [Lyons and Thorne, 1973] and is not in itself a new idea. Nonetheless, the paper would still be of interest to the community if the diffusion modeling were made more convincing and I would recommend that the authors consider a resubmission after considering the comments below.

Thank you. As described above, hopefully with the knowledge that the modelling does in fact include the wave-particle losses that the referee identified as important, and the additional emphasis that energy diffusion effects are negligible at ultra-relativistic energies, the referee will find our results more convincing. As such, we hope that they can now recommend our work for publication in Nature Communications.

Specific comments for the authors to address

P2 There has been substantial improvement in our quantitative understanding of the relative importance of various processes that affect the radiation belts since the paper of Reeves et al. [2013]. The statement that the relative importance of these processes "is not well understood" is not correct.

We agree with the assessment that the Van Allen Probes have indeed improved our understanding of the different processes which are the building blocks which must explain the observed wide range of belt dynamics. Nonetheless, we think it remains the case that their relative importance under different conditions is still a work in progress and is (at least not yet) still not well-understood. For example, the Reeves et al. (2013) paper relates to a peak in phase space density for a specific storm, but as yet (to our knowledge) there are as yet no published statistics of the prevalence of such peaked PSD distributions for an ensemble of storms. Certainly it is well-known statistically, for example, that the response to ICMEs and fast streams/CIRs is not the same. That observation likely requires an understanding of the relative importance and/or nature of the action of different modes of loss and acceleration in order to generate the observed response.

However, we do take the referees point. Hence the revised text states that “However, how these processes act to collectively produce the observed morphology, and the relative importance of multiple competing acceleration and loss processes under different solar wind driving conditions, remains relatively poorly understood⁵.” We believe this reflects the appropriate level of understanding at this time.

P2 The apparent barrier only applies to the highly relativistic population. Lower energy electrons are able to be transported into lower L during solar activity and fill the gap between the inner and outer radiation belts [see Ma et al., 2016]. This is an important distinction that needs to be discussed in the concept of energy dependent radial transport.

We completely agree.

There will be additional physical processes operative at lower energies; and indeed at these lower energies the barrier is not observed. Of course such lower energy particles appear to obey different physics and are not the focus of the current paper. In the title of our paper, the abstract

text, in the introduction, and in the descriptions throughout the manuscript, we make it clear that we are focussing on, and simulate, the response at the ultra-relativistic energies where the barrier phenomena is observed. However, the referee's suggestion that this distinction be made explicit in the paper is a good one, and we now reference the observational paper by Reeves et al. (2016) as well as the modelling paper on the March 2013 storm by Ripoll et al. (2016). In addition, as suggested by the referee we also cite the Ma et al., (2016) modeling paper also on the March 2013 storm.

Section on Results: In Fig 1 there appears to be little difference in the modeling results using the radial diffusion rates of B&A and Ozeke, and neither of the two simulations are able to explain the apparently slow (months) inward diffusion of 4.2 and 5.6 MeV electrons following a major storm. This suggests that neither radial diffusion rate is correct during less active times following a storm.

That is probably true. This might be because of the relative lack of attention or larger errors in relation to assessing the low activity transport rates, and/or relating to the challenges associated with specifying month-timescale and/or low-L transport. In relation to the Ozeke diffusion coefficients at low-L, the mapping from the ground magnetometer ULF wave power to the equatorial electric fields has some uncertainties especially at $L < \sim 4$. Given the B&A L-dependence is done using a crude extrapolation between observations from two L-shells, there are potentially large errors in the B&A empirical specification as well. However, to our knowledge these are the best empirical models which currently exist. Future studies in the community may be able to address this with the development of even better transport coefficient specifications.

Setting an outer boundary condition from observed flux measurements at L=6 is not appropriate since this ignores the possibility of any local acceleration source, which has been shown to produce peaks in phase space density between $L = 4$ to 5 during storms.

As discussed above such energy diffusion is negligible at ultra-relativistic energies, such that our approach is valid.

When such peaks are formed the radial diffusion direction is outward to regions of lower PSD near the boundary [Turner et al., 2014]. This leads to the concept of magnetopause shadowing as a loss process for the radiation belts, which has been addressed in numerous papers [e.g., Shpritz et al., 2006] prior to the referenced paper of Mann et al., [2016]. This is but one example of the biased choice of references used in the manuscript.

The two cited papers relate to outward radial diffusion [Shprits], and the application of very fast outwards ULF wave radial diffusion for the creation of the third radiation belt [Mann], respectively. Of course, reductions of PSD observed at the outer boundary by Van Allen Probes do create shadowing losses when imposed in our model – but this *loss* cannot explain the depth of penetration of the belt *enhancement* to the fixed location of the barrier addressed here. Nonetheless we have also added some new references in relation to the physics of magnetopause shadowing.

In relation to outer radiation belt loss due to magnetopause shadowing, the revised manuscript now states:

“In addition, outward ULF radial transport in combination with magnetopause shadowing can also result in outer radiation belt electron loss^{11, 12, 13.}” Referencing the work of:

11. West, H., Buck, R. & Walton, J. Shadowing of electron azimuthal-drift motions near the noon magnetopause. *Nature* **240**, 6-7 (1972).

12. Turner, D. L., Shprits, Y., Hartinger, M. & Angelopoulos, V. Explaining sudden losses of outer radiation belt electrons during geomagnetic storms. *Nature Physics* **8**, 208 (2012).

13. Ozeke, L. G. *et al.* Modeling cross L shell impacts of magnetopause shadowing and ULF wave radial diffusion in the Van Allen belts. *Geophys. Res. Lett.* **41**, 6556-6562 (2014).

Another example of misleading referencing occurs in the discussion of losses due to EMIC waves. Here the paper of Usanova et al [20014] is cited to explain the fact that EMIC waves are only responsible for scattering of electrons at low pitch angles around the loss cone. But this was clearly shown in much earlier studies [e.g., Lyons and Thorne, 1972; Li et al., 2007].

The cited papers [e.g., Lyons and Thorne, 1972; Li et al., 2007] are both modelling papers and contain no data or observations. The Usanova et al. paper presents (to our knowledge the first) observations which show the effect of preferentially low PA EMIC-related loss clearly – of course in agreement with prior theory, as well as modelling building on this which is included in the Usanova et al. paper. Nonetheless to make it clear that the Usanova et al. work is observational and draws on earlier modelling work we have modified this sentence and added these two Lyons and Thorne, 1972; Li et al., 2007 references as Refs^{34,35}. Note also that the Li et al. (2007) paper is cited in Usanova et al.

The text now reads: “However, observations presented by Usanova et al.³³ show how the action of EMIC waves alone only impacts low equatorial pitch angle particles (where pitch angle is the angle between particle velocity and the background magnetic field) such that these waves acting alone are not expected to be able to deplete the core of the distribution^{34,35}.”

Section on "Does the Barrier exist".

In this section the authors use their model for radial diffusion to simulate the potential effect of strong solar wind driving over an extended period of time to show that radial diffusion could bring relativistic electrons into lower L if high levels of activity ($K_p > 6$) were sustained for tens of days. However, this is very unrealistic and has never been observed in Nature. Consequently, I would suggest removing this section or at least moving it into supplementary material.

The Figure referred to by the referee shows example results for a range of constant Kp values, each illustrating situations where the belts are initially empty at all L-shells; therefore, the simulations illustrate the approximate activity dependence of the transport rates on the timescale of 10 days. As we show in detail in our manuscript the actual transport rate additionally depends also on df/dL , such that such simulation results are valuable.

We agree with the reviewer that sustained activity of $Kp > 6$ for ten(s) days in the terrestrial system is not typical; however, it is not unrealistic and has been observed in nature. Figure 5 illustrates that such a long period of $Kp > 6$ activity is not required for the outer radiation belt electrons to be radially transported inward onto lower L-shells below $L \sim 2.8$. Specifically, Figure 5 shows that at $L=3$ the inward transport time for a sustained activity of $Kp=6$ is ~ 2 days (bluey/green color) and for $Kp=7.7$ it is around 0.5 days (blue color). We have added new text to the manuscript which highlights that for the 2003 Halloween storm where the slot was penetrated Kp was >6 for two intervals of 24 hours, and reached maximum values of $Kp=9$ (see the plot included in our reply below). In addition, during the November 2004 storm a sustained value of $Kp > 6$ lasted for 36 hours.

Other non-terrestrial systems may also show different timescales of activity than the terrestrial case.

However, overall the problem we address is well-posed and we remain of the view that Figure 5 is helpful to the reader to understand the impact of sustained activity on the inward radial transport of radiation belt electrons including df/dL effects.

The length of the paper is also well within the limits for Nature Communications in its new open access format.

Section on March 2015 storm

I agree with the authors that rapid loss from VLF transmitters is not needed to explain the sharp inner gradient in PSD formed during this particular storm.

Thank you. We of course agree – but this is in direct contrast to the published results of Foster et al. (2016) discussed in the paper. As a result, we believe our results are an important contribution to this scientific debate and should be published.

But here again the simulations performed to support this do not contain any realistic losses by wave scattering or the effect of local energy diffusion outside the plasmopause during interactions with chorus waves. It would be much more convincing if such effects were included.

As explained above, our simulations do already include realistic losses as sensibly requested by the reviewer; local energy diffusion effects can be neglected at the ultra-relativistic energies examined here. With this explanation, we hope that the referee will be now convinced by our results and able to recommend our paper for publication.

We thank Referee #3 for the careful reading of our manuscript. We have revised the paper taking into account the comments from the reviewers in our revision. In our responses below, the original comments from Referee #3 are presented in **bold typeface** and our replies in regular font.

We hope that with these changes the referee can now recommend our paper for publication in Nature Communications.

Reviewer #3

This paper presents very interesting results showing the so-called ‘impenetrable barrier’ at $L=2.8$ (Baker et al.) for highly relativistic electrons can be explained by the properties of radial diffusion due to ULF waves, in combination with the typical time-scales for large storms.

Thank you.

The authors do a careful job of showing the effects of strong solar wind driving (as parameterized by Kp) on the timescale for relativistic electrons to reach different L values. They show that in fact, for larger driving, ultra-relativistic electrons can penetrate inside $L=2.8$ and that there is nothing special about that location. As the authors point out, a number of other researchers have shown that during very large storms electrons are seen inside this location.

Thank you.

Overall, we are glad the referee thinks “*This paper presents very interesting results showing the so-called ‘impenetrable barrier’ at $L=2.8$ (Baker et al.) for highly relativistic electrons can be explained by the properties of radial diffusion due to ULF waves, in combination with the typical time-scales for large storms.*” As suggested by the reviewer a number of minor changes have been made in the revised manuscript these are outlined below.

Issues to be addressed before publication:

The referencing of other work on ULF waves impact on radiation belts is not complete.

Although loss due to whistler-mode waves is addressed, other loss mechanisms such as magnetopause shadowing are not addressed.

More referencing to other work on the impacts of ULF waves and magnetopause shadowing on radiation belt dynamics is included in the revised manuscript. In relation to ULF wave diffusion we now reference:

Schulz, M. & Lanzerotti, L. J. in *Particle diffusion in the radiation belts* (Springer Science & Business Media, 2012).

In relation to outer radiation belt loss the revised manuscript now states:

“In addition, outward ULF radial transport in combination with magnetopause shadowing can also result in outer radiation belt electron loss^{11, 12, 13.}” Referencing the work of:

11. West, H., Buck, R. & Walton, J. Shadowing of electron azimuthal-drift motions near the noon magnetopause. *Nature* **240**, 6-7 (1972).

12. Turner, D. L., Shprits, Y., Hartinger, M. & Angelopoulos, V. Explaining sudden losses of outer radiation belt electrons during geomagnetic storms. *Nature Physics* **8**, 208 (2012).

13. Ozeke, L. G. *et al.* Modeling cross L shell impacts of magnetopause shadowing and ULF wave radial diffusion in the Van Allen belts. *Geophys. Res. Lett.* **41**, 6556-6562 (2014).

Figure 1 should be redrafted so that each energy is full page width. It is not possible to see the relationship between the plasmopause location and the electron fluxes on the scale shown.

The figure caption is also confusing and should be .

This is true of several other figure captions, which do not stand on their own. One can't interpret the figure from the captions alone.

As suggested Figure 1 has been redrafted so that each energy is full page width. In addition, the black dotted line representing the plasmopause has been replaced with a solid white line which is easier to see. The Figure captions have also been slightly revised for clarity, as requested by the reviewer.

The authors do not need to continually emphasize the term used in the Baker et al. (and following articles) paper - "impenetrable barrier" - through use of italics and quotation marks.

Finally, as requested by the reviewer, the repeated use of quotation marks in relation to the “impenetrable barrier” has been removed from the revised manuscript.

Reviewers' comments:

Reviewer #1 (Remarks to the Author):

I consider that the authors have adequately responded to the reviewers' comments and have appropriately modified their manuscript. I therefore regard this paper as suitable for publication.

Reviewer #2 (Remarks to the Author):

The authors have made an attempt to answer some of the comments in my first round of reviews. However, there are still several instances where the manuscript is misleading and these need to be corrected before I would recommend publication.

1) Stochastic energy diffusion at ultra-relativistic energies is not negligible as the authors claim. It is just less efficient than at lower energies. The cited paper by Shprits et al [2013] is incorrect on this issue. As demonstrated by Thorne et al. [2013] energy diffusion to ultra-relativistic energies can occur within the duration of strong geomagnetic activity ($< \text{day}$) for certain storms. This has now been confirmed by several other simulations and the results of the local acceleration near $L \sim 4$ leads to observed peaks in phase space density just outside the plasmopause. In the region of negative phase space density gradient outside the peaks, radial diffusion would carry the energetic electron population outwards towards the magnetopause where they are ultimately lost. The simulations performed in the paper under review do not take this fundamentally important process into account. Instead they assume that electron diffuse inwards from the magnetopause throughout the storm, which is clearly not correct based on numerous observational studies of the local peaks in phase space density in the heart of the radiation belts.

2) The physical processes operating on the lower energy radiation belt population ($< 1 \text{ MeV}$) is not substantially different from that at ultra-relativistic energies, it is simply a matter of the time scales of operating loss and acceleration processes. Although the authors have now acknowledged that there is no apparent barrier to the access of lower energy electrons to the inner radiation belts, no explanation is offered for this important difference. One potential reason for this may be due to the gradients in electron phase space density, which always tends to remain positive out to the magnetopause at lower values of μ . But since the authors do not show profiles of phase space density it is difficult for the reader to judge. Presumably they have this information since the radial diffusion equation is solved for phase space density at constant μ .

Reviewer #3 (Remarks to the Author):

The revised paper has addressed the issues raised in my initial report. The features in the figures are easier to see; the captions are now much clearer. The paper is publishable.

The authors have added references on shadowing, but only to their own work. I suggest that they add:

Hudson, M. K., D. N. Baker, J. Goldstein, B. T. Kress, J. Paral, F. R. Toffoletto, and M. Wiltberger (2014), Simulated magnetopause losses and Van Allen Probe flux dropouts, *Geophys. Res. Lett.*, 41, 1113–1118, doi:10.1002/2014GL059222.

Similarly on 'other loss mechanisms' they should add:

Millan, R. M., and R. M. Thorne (2007), Review of radiation belt relativistic electron losses, *J. Atmos. Sol. Terr. Phys.*, 69, 362–377.

Reply to Referees: Second Review

Nature Communications paper "Explaining the Apparent Impenetrable Barrier to Ultra-relativistic Electrons in the Outer Van Allen Belt" by Ozeke et al.

We thank the reviewers again for their careful reading of the revised version of our paper and for their constructive comments on the manuscript. We believe that we have now addressed the final small number of comments from the reviewers, and that with our latest revision Referees #2 will join Referees #1 and #3 in now recommending our paper for publication in Nature Communications.

We address the referees detailed comments below. The referees' comments are provided in standard font, and our responses below are provided in bold type.

Reviewers' comments:

Reviewer #1 (Remarks to the Author):

I consider that the authors have adequately responded to the reviewers' comments and have appropriately modified their manuscript. I therefore regard this paper as suitable for publication.

Thank you, and we also now look forward to publication of our paper.

Reviewer #2 (Remarks to the Author):

The authors have made an attempt to answer some of the comments in my first round of reviews.

However, there are still several instances where the manuscript is misleading and these need to be corrected before I would recommend publication.

We took the referees comments extremely seriously in our previous revision, and we are pleased that there appear to only be two remaining concerns from Referee #2 both of which we address below.

We are not sure what specific elements in the revised paper that the referee considers to be "misleading". Nonetheless, we take the Referee's comments very seriously and address them in detail below.

1) Stochastic energy diffusion at ultra-relativistic energies is not negligible as the authors claim. It is just less efficient than at lower energies. The cited paper by Shprits et al [2013] is incorrect on this issue.

This appears to be a statement of the referee's opinion. We however very much agree that these effects are weaker at ultra-relativistic energies, and indeed are likely to be very weak on L-shells inside the plasmopause and even weaker in the proximity of the barrier at L~2.8 where lower band chorus waves are extremely rare.

The crux of the issue is whether the efficiency of this energy diffusion at ultra-relativistic energies and at low-L is slow enough to be neglected, as argued by Shprits et al. (Nature Physics, 2013). In particular, for the purposes of this paper, the issue is specifically whether the location of the apparently impenetrable barrier can be explained in terms of the low-L radial transport physics.

We contend the answer to both of these questions is yes, and our simulations support this conclusion.

We emphasise that we are not claiming that stochastic energy diffusion is never important in the belts, but that it does not control the location of the barrier. As the referee states, such acceleration is less

efficient at the ultra-relativistic energies associated with the barrier, and is even less significant at low-L such as at $L \sim 2.8$ where the barrier occurs.

We note that in our simulations we impose the observed dynamics of the phase space density from close to the apogee of the Van Allen Probes either at $L=6$ (long duration simulation) or at $L^*=5$ (March 2015 storm) as the basis for transport to the barrier. We make no claims about what controls the observed variation of flux at that outer boundary. Indeed, it is quite possible that stochastic energy diffusion and/or the action of local acceleration could play a role in creating the observed PSD dynamics at the outer edge of the domain. However, this does not invalidate the results from our simulations nor our conclusions.

We now make this clear with new text on page 13 where we state: *“There could also be impacts from the action of chorus wave acceleration, as described for example by Thorne et al.⁶. These effects are expected to often be relatively weak at ultra-relativistic energies³⁹, and if they primarily act close to the outer boundary of our simulations the inward ULF wave transport of this additional source of flux will be captured in our model.”*

On page 29 we also added the following text to additionally clarify this in the Methods section: *“Even if local acceleration rapidly creates an additional source for electrons around $L=5$ close to the edge of our simulations, as argued by Thorne et al.⁶, the inward transport of such sources to the apparent barrier at $L \sim 2.8$ will also be captured in our simulations.”*

We note here that the focus of our paper is on the inward transport of flux to the location of the apparent barrier at $L \sim 2.8$. This paper is not focussed on the debate about the nature of radiation belt acceleration.

We note further that neither Referee #1 nor Referee #3 share this concern.

As demonstrated by Thorne et al. [2013] energy diffusion to ultra-relativistic energies can occur within the duration of strong geomagnetic activity ($< \text{day}$) for certain storms. This has now been confirmed by several other simulations and the results of the local acceleration near $L \sim 4$ leads to observed peaks in phase space density just outside the plasmopause.

We reiterate that we not claiming that there can never be any acceleration by chorus waves at ultra-relativistic energies, but argue that our simulations show that it is the very steep and strong L-shell and activity dependence of the radial transport rates at lower L which control the eventual location of the impenetrable barrier at $L \sim 2.8$.

We note also that the comparison to observations presented in Thorne et al. (2013) [their Figure 3] is done at $L=5$ so in fact is very close to the outer boundary of the barrier simulations presented here. Hence that paper does not invalidate our conclusions. As per the new text in our paper (copied above) we also now explicitly highlight in the text how such new sources of flux are effectively incorporated into the inward transport in our model through the use of observed flux at the outer simulation boundary.

In the region of negative phase space density gradient outside the peaks, radial diffusion would carry the energetic electron population outwards towards the magnetopause where they are ultimately lost.

The simulations performed in the paper under review do not take this fundamentally important process into account.

Our simulations include both inwards and outwards radial diffusion. In our model these arise from dynamical phase space density variations at the outer simulation boundary – and such variations especially increases in PSD there could occur from the action of the local acceleration that the referee highlights.

Instead they assume that electron diffuse inwards from the magnetopause throughout the storm, which is clearly not correct based on numerous observational studies of the local peaks in phase space density in the heart of the radiation belts.

This is not a correct statement. Our simulations examine transport to the barrier from a fixed location which is inwards of the magnetopause. The simulations include both inwards and outwards radial transport, albeit imposed from dynamical flux variations at the outer boundary simulation boundary at e.g., $L^* = 5$. Such observed flux variations could come from both the energy diffusion highlighted by the referee (perhaps increasing PSD at the boundary) as well as impacts there from magnetopause shadowing (reducing the PSD at the boundary). Our paper simply contends that the very strong L-shell and activity dependence of the radial diffusion coefficients from the outer simulation boundary down to low L explains the creation and location of the apparent barrier.

Overall, our simulations represent a well-posed scientific problem and we use a well-defined methodology and which produces an explanation for the location of the barrier in agreement with observations.

Note that we also show how the barrier is well-reproduced in extremely long-term simulation runs spanning over 13 months, and does so without any data assimilation inside the simulation domain. The only inputs are the dynamical flux at the outer boundary, which can include the effects of chorus whose importance is highlighted by the referee, as a result of our specifying the accurate rates of inward (and outward) transport.

We acknowledge that our simulation results do not always perfectly recreate the absolute magnitude of the flux changes inside the domain, and perhaps this is a signature of some weak and slow missing losses or even additional local (energy diffusion) acceleration process at ultra-relativistic energies. However, that is not the focus of our paper.

None of this affects our conclusion that time- and L-dependent variations in radial transport rates at low-L explain the very specific feature of the location of the impenetrable barrier. Referees #1 and #3 agree fully with our interpretation.

2) The physical processes operating on the lower energy radiation belt population (<1 MeV) is not substantially different from that at ultra-relativistic energies, it is simply a matter of the time scales of operating loss and acceleration processes.

We agree, but this is a rather different problem and not the focus of this paper.

For example, slow and inefficient processes which act at ultra-relativistic energies at low-L can presumably be ignored as they will not be dominant; conversely if these processes are stronger and faster at lower energies then they should be included in studies of lower energy particle dynamics. We do not address the lower energy (<1 MeV) population here, since the impenetrable barrier is an ultra-relativistic feature.

Overall, the sharp barrier is only observed at ultra-relativistic energies and has yet to be explained. We explain it here.

Although the authors have now acknowledged that there is no apparent barrier to the access of lower energy electrons to the inner radiation belts, no explanation is offered for this important difference.

This is the same comment the referee raised in their previous review and which we addressed by including additional references which not only describe this lower energy phenomena but also how it is impacted by different physics (Refs. 20, 21, 22).

We also draw the referee's attention to the recent paper by Ripoll et al. (2017) which studies the so-called "S" shape of the energy-L dependence of lower energy (not ultra-relativistic) particles (see also Reeves et al. (2016); our Ref. 20), and emphasises a strong energy dependence to hiss losses at these

lower energies as the explanation for lower energy electron penetration deep into the slot. We also now cite the Ripoll et al. (2017) paper in our manuscript as Ref. 11.

As shown for example by Ripoll et al. (2017), at lower non-ultra-relativistic energies other energy dependences of the hiss loss process can become important; and that requires different physics, and is a different problem. Hence whilst such lower energy dynamics are interesting they are not related to the subject at hand – i.e., the creation of the ultra-relativistic barrier.

Despite the advances in the Ripoll et al. (2017) paper, the question of the energy dependence of slot penetration is one which continues to be at the forefront of current research.

Overall, whilst the lower energy dynamics are certainly interesting they represent a different problem, governed by rather different physics, and hence are not relevant to the topic of this publication.

One potential reason for this may be due to the gradients in electron phase space density, which always tends to remain positive out to the magnetopause at lower values of μ . But since the authors do not show profiles of phase space density it is difficult for the reader to judge. Presumably they have this information since the radial diffusion equation is solved for phase space density at constant μ .

We agree the lower μ dynamics are interesting, but we reiterate that these lower energy penetrations involve some different physics (cf. the very recent Ripoll et al. (2017) paper published in July 2017) and are not the subject of this paper.

Reeves, Geoffrey D., et al. "Energy-dependent dynamics of keV to MeV electrons in the inner zone, outer zone, and slot regions." Journal of Geophysical Research: Space Physics 121.1 (2016): 397-412. [included as Ref. 20 in our manuscript]

Ripoll, J.F., Santolik, O., Reeves, G., Kurth, W.S., Denton, M., Loridan, V., Thaller, S., Kletzing, C.A. and Turner, D.L., 2017. Effects of whistler mode hiss waves in March 2013. Journal of Geophysical Research: Space Physics. [now included as Ref. 11 in our manuscript]

Reviewer #3 (Remarks to the Author):

The revised paper has addressed the issues raised in my initial report. The features in the figures are easier to see; the captions are now much clearer. The paper is publishable.

Thank you, and we also now look forward to publication of our paper.

The authors have added references on shadowing, but only to their own work. I suggest that they add:

Hudson, M. K., D. N. Baker, J. Goldstein, B. T. Kress, J. Paral, F. R. Toffoletto, and M. Wiltberger (2014), Simulated magnetopause losses and Van Allen Probe flux dropouts, Geophys. Res. Lett., 41, 1113–1118, doi:10.1002/2014GL059222.

Similarly on 'other loss mechanisms' they should add:

Millan, R. M., and R. M. Thorne (2007), Review of radiation belt relativistic electron losses, J. Atmos. Sol. Terr. Phys., 69, 362–377.

These two references have now been added, see Ref. 16 and 12.

Reviewers' comments:

Reviewer #2 (Remarks to the Author):

First let me state that I agree with the authors that the "impenetrable barrier" is not caused by a sudden increase in loss of relativistic electrons at lower L. I also agree that the rate of inward radial diffusion becomes progressively slower at low L and that this is a main contributor to the inability of relativistic electrons to gain access to the region inside $L \sim 2.8$ during storms that occurred over the recent Van Allen probes era. However, one should note that there are exceptions to this such as the unique March 1991 event where >16 MeV electrons were injected well inside the location of the so-called "impenetrable barrier". My principal objections to the paper under review is that the authors dismiss or minimize the effects of local acceleration by chorus emissions and also do not include realistic loss processes in their 1D radial diffusion modeling.

Specifically, the discussion on p13 line 240 -242 gives the impression that local acceleration by chorus is ineffective at relativistic energies citing again the flawed paper by Shprits et al. [2013]. I have personally spoken with Shprits about this and he now agrees that local acceleration to relativistic energies does indeed occur. The time scale for this is typically on the order of a day which is slower than that at lower energies but still sufficient to explain the increase in relativistic flux during most storm conditions. Following the paper by Thorne et al [2013] modeling by numerous other groups have confirmed the importance of the local acceleration process and the build-up of peaks in phase space density of relativistic electron in the region just outside the stormtime plasmopause during numerous storms in the Van Allen era. These peaks in PSD typically occur near $L \sim 4-5$ and would cause outward radial diffusion at all locations exterior to the peak. The author's misleading statement that local acceleration is "typically very weak at ultrarelativistic energies" is repeated again in the Methods section lines 506-507. I cannot recommend publication with such erroneous statements in the paper.

Another issue I have with the paper under review is the absence of any realistic loss in their modeling. This is clearly evident in the simulations shown in Figure 1.

Reply to Referees: Third Review

Nature Communications paper "Explaining the Apparent Impenetrable Barrier to Ultra-relativistic Electrons in the Outer Van Allen Belt" by Ozeke et al.

We thank the reviewer again for their careful reading of the re-revised version of our paper. We have addressed all of the small number of final comments. We hope that with our latest revision Referee #2 will join Referees #1 and #3 in recommending our paper for publication in Nature Communications.

We address the referees detailed comments below. The referees' comments are provided in standard font, and our responses below are provided in bold type.

Reviewers' comments:

Reviewer #2 (Remarks to the Author):

First let me state that I agree with the authors that the "impenetrable barrier" is not caused by a sudden increase in loss of relativistic electrons at lower L. I also agree that the rate of inward radial diffusion becomes progressively slower at low L and that this is a main contributor to the inability of relativistic electrons to gain access to the region inside $L \sim 2.8$ during storms that occurred over the recent Van Allen probes era.

Thank you very much; we are very pleased that the referee agrees with our conclusions.

The transport process the referee describes is the main point of our paper and we contend that these are new results which deserve to be published in the literature.

Our results explain the location of the barrier and contrast and correct alternative interpretations which have been previously presented in the literature. We are also pleased that the referee agrees with us that these other alternative explanations are not correct.

Since our explanation has not been published elsewhere, we believe that this is the strongest argument in favour of the publication of our paper.

However, one should note that there are exceptions to this such as the unique March 1991 event where >16 MeV electrons were injected well inside the location of the so-called "impenetrable barrier".

We of course agree.

As the Referee no doubt knows-well, such super-relativistic (in this case > 16 MeV) acceleration happens in only minutes as a result of a shock impact (e.g., Li et al., 1993). Of course these rapid shock acceleration events are a different phenomena and are also rather rare. For example, there has not been an event as strong and as clear as the March 1991 event for the entirety of the Van Allen Probes era thus far (since 2012). As such they are not related to the barrier phenomena as reported as a long-lasting phenomenon in the Van Allen Probes era.

On lines 275 – 278 our manuscript also explicitly acknowledges how other extreme superstorm events might penetrate the so-called impenetrable barrier, for example as in relation to the slot penetration during the Halloween 2003 superstorm.

My principal objections to the paper under review is that the authors dismiss or minimize the effects of local acceleration by chorus emissions and also do not include realistic loss processes in their 1D radial diffusion modeling.

As we justify below, we argue that these objections are not valid and that neither criticism affects the validity of our conclusions. We hope that in considering our justification and responses below that the referee will agree and now recommend our paper for publication.

Local acceleration effects highlighted by the reviewer typically occur around $L^* \sim 5$ (cf. eg Thorne et al., 2013), close to the outer boundary of our simulations. Since our model ingests the observed flux variations there, our model will in fact include effects related to the inward transport of such new flux to the barrier.

This occurs regardless of the physical processes which change the flux at the outer boundary of our model – including local acceleration. Therefore the boundary conditions of our model would include such local acceleration effects if they are active there by ingesting any resulting increases in observed fluxes at $L^* \sim 5$.

We now make the potential importance of this, and its relation to our simulations, explicit in the manuscript at lines 239-244 where we state:

“There could also be impacts from the action of chorus wave acceleration, as described for example by Thorne et al.6. While chorus waves may play an important role in the acceleration of electrons at relativistic energies 39 40, at the ultra-relativistic energies ($\gtrsim 2$ MeV) examined here the effects are expected to often be relatively weak 41. Moreover, if additional chorus acceleration also primarily acts close to the outer boundary of our simulations the inward ULF wave transport of this additional source of flux will be captured in our model.”

As for the comment in relation to realistic loss models, our model includes losses from hiss and chorus scattering based on the best current empirical models in the literature (details at Methods section (Line 484 onwards, e.g., explicitly at Lines 501-505)). Such data-driven models are as realistic as is currently available.

Specifically, the discussion on p13 line 240 -242 gives the impression that local acceleration by chorus is ineffective at relativistic energies citing again the flawed paper by Shprits et al. [2013]. (*emphasis added ours*).

We have no intention of inadvertently implying that such local acceleration effects are unimportant at relativistic energies, and we agree such local acceleration effects could indeed be much more important at lower energies below ultra-relativistic. But the barrier is not observed at those lower energies and hence such issues are not related to this manuscript.

Nonetheless the new text we have added at lines 239-244 (copied above) also calls out explicitly the importance of possible impacts of local acceleration at lower relativistic energies.

The bottom line is that our modelling of transport to the barrier remains valid.

I have personally spoken with Shprits about this and he now agrees that local acceleration to relativistic energies does indeed occur. The time scale for this is typically on the order of a day which is slower than that at lower energies but still sufficient to explain the increase in relativistic flux during most storm conditions. (*emphasis added ours*)

We have no comment to make about the referee’s private dialogue with Shprits on the nature of the dominant acceleration processes active at relativistic energies. Particles at such energies are not the focus of our paper.

Instead, our paper focusses on modelling the inward transport from $L^* \sim 5$ to create the ultra-relativistic barrier at $L \sim 2.8$. We are also unaware of any studies which show that local acceleration might be significant at the low L-shells ($L \sim 2.8$) close to the barrier. Hence as described above our analysis and conclusions remain valid.

Even if local acceleration is important at $L^* \sim 5$, as the referee contends, we now describe explicitly in the text how our modelling would include it since it ingests the variations of observed fluxes at the simulation outer boundary at $L^* \sim 5$.

Following the paper by Thorne et al [2013] modeling by numerous other groups have confirmed the importance of the local acceleration process and the build-up of peaks in phase space density of *relativistic* electron in the region just outside the stormtime plasmapause during numerous storms in the Van Allen era. (*emphasis added ours*).

We do not disagree with this statement. However, we reiterate as per our reply above that our paper is not concerned with *relativistic* energies, nor indeed with the debate about dominant acceleration processes. Instead we are interested in the transport of source fluxes from $L^* \sim 5$ to create a barrier to ultra-relativistic electrons at the inner edge of the Van Allen belts at $L \sim 2.8$. Such sources around $L^* \sim 5$ are included in our model courtesy of the boundary conditions which are imposed there.

These peaks in PSD typically occur near $L \sim 4-5$ and would cause outward radial diffusion at all locations exterior to the peak.

We agree. Certainly the referee is correct that there would be outward transport from a such peak in PSD at $L \sim 5$. However, our simulations address the *inward* transport to the barrier from observed source fluxes in this location – however such source fluxes are created.

Therefore none of the referees statements above invalidate our conclusions about the physics which explains the barrier location.

The author's misleading statement that local acceleration is "typically very weak at ultrarelativistic energies" is repeated again in the Methods section lines 506-507. I cannot recommend publication with such erroneous statements in the paper.

Consistent with lines 239-244, at line 507 we have also modified the text to replace the word "very" with "relatively", and again explicitly stated that local acceleration may be more important at relativistic energies. It now reads "...typically relatively weak at ultra-relativistic energies as compared to relativistic energies". This statement now makes the relative comparison between relativistic and ultra-relativistic energies, this wording being entirely consistent with the Referees own written characterisation of this topic from the previous (second) round of reviewing.

Another issue I have with the paper under review is the absence of any realistic loss in their modeling. This is clearly evident in the simulations shown in Figure 1.

This statement is not accurate. As we explained in our earlier two replies to this referee, and which we briefly repeat here, data-derived hiss and chorus loss models which are obtained from wave observations are used in our modelling. These have been used in numerous other simulation studies in recent papers, and to our knowledge are the most realistic empirical models available. The relevant published papers and the definition of our approach are given in our manuscript at lines 501-505.

We also explicitly acknowledge and discuss at lines 209-244 that the agreement between our simulations and the data in terms of absolute flux magnitude is not always perfect, and we openly discuss for the reader some possible reasons for this discrepancy. Other simulation studies such as that by Drozdov et al. (2015) have noted a similar issue with modelling absolute flux. Indeed, the results from our long-duration simulations shown in Figure 1 compare extremely favourably with the fidelity of other recently published long-term simulation results (for example that of Drozdov et al. (2017) and which was published in August 2017 during the period whilst this paper has been under review).

The referees comment does not in any way invalidate our conclusions about transport to the barrier. It is certainly not the case that our simulations do not include "any realistic loss modelling".

References: Drozdov, A. Y., et al. (2015), Energetic, relativistic, and ultrarelativistic electrons: Comparison of long-term VERB code simulations with Van Allen Probes measurements. *J. Geophys. Res. Space Physics*, 120, 3574–3587. doi: 10.1002/2014JA020637.

Drozdov, A. Y., et al. "EMIC wave parameterization in the long-term VERB code simulation." *Journal of Geophysical Research: Space Physics* (2017).

REVIEWERS' COMMENTS:

Reviewer #2 (Remarks to the Author):

If the authors had shown the changes in PSD during the simulated event rather than flux at a single energy it would be much easier for the reader to understand how much of the radial diffusion was inward or outward and I encourage them to do this in the future. As for the loss process at relativistic electrons during such events scattering by EMIC waves near the loss cone in combination with scattering by hiss or chorus at larger pitch angles has been demonstrated in several previous studies to be most effective. This paper under review makes no attempt to model such combined loss, hence my comment that the loss included is not realistic. In fact a 2D simulation would be need to treat this properly. However, as the authors state this is not the main point of their study. My main criticism in the last round of review related to the way that local acceleration by chorus waves was describes and implied to be of little importance. But with the additional clarifications on the role of local acceleration I have no further comments on the manuscript and am willing to recommend publication.